# A neural circuit for context-dependent multimodal signaling in *Drosophila*

Elsa Steinfath[1,3], Afshin Khalili[1,3], Melanie Stenger[1,2], Bjarne L. Schultze [1,2], Sarath Ravindran Nair[1], Kimia Alizadeh[1] & Jan Clemens [1,2] ✉

Many animals produce multimodal displays that combine acoustic, visual, or vibratory signals, yet the neural mechanisms coordinating these behaviors remain unclear. Using *Drosophila* courtship as a model, we reveal how a single neural circuit integrates sensory cues and motivational state to orchestrate multimodal signaling. Male flies produce both air-borne song and substrate-borne vibrations during courtship, but in distinct, largely non-overlapping contexts. We demonstrate that the same brain neurons that drive song also control vibrations through separate pre-motor pathways, with cell-type specific dynamics. This shared circuit coordinates multimodal displays with locomotion, ensuring vibrations are produced only when they can effectively reach the female. The circuit employs shared motifs—recurrence and mutual inhibition—that enable dynamic control of multimodal signals by external cues and internal state. A computational model confirms that these motifs are sufficient to explain the observed behavioral dynamics. Our findings illustrate how simple neural circuit elements can be combined to select and coordinate complex multimodal behaviors.

Social communication is inherently multimodal. During conversations, we are not mere loudspeakers that emit speech but coordinate our words with dynamical facial expressions and other body gestures. Gestures produced in congruence with speech rhythms can improve comprehension[1,2] whereas reducing multimodality, as in phone calls, can impair it[3,4]. Multimodal communication is not unique to humans[5,6] but also prevalent in other animals. For instance, monkeys[7], birds[8], frogs[9], or grasshoppers[10] combine acoustic signals with visual displays[7,11,12], while many insects combine sound with substrate-borne vibrations[13–18]. Effective multimodal communication requires the production of the appropriate sequence or combination of signals contingent upon the context, for example, coordinating movements with a dance partner[19,20].

Due to the multifaceted nature of multimodal signaling, the underlying brain circuits have mainly been studied by isolating single components of this behavior[7,21–26], but their contribution to the coordination of multimodal signals is not well understood. Moreover, the

mechanisms by which these circuits integrate external cues for context-appropriate signaling[27,28] and coordinate signaling with ongoing behaviors such as respiration and locomotion transmission are poorly understood[29,30]. At one extreme, parallel circuits could independently integrate the specific external cues required to trigger different behaviors[5]. Alternatively, a single integrated circuit could trigger multiple behaviors and signal coordination arises from the interaction between external sensory inputs, internal motivational state, and circuit dynamics[31–33].

Here, we address the issue of multimodal signaling in *Drosophila melanogaster*. During courtship, male flies chase females while producing both air-borne song and substrate-borne vibration[34,35]. Song is produced by extending and fluttering one wing, resulting in two distinct modes: a sine song characterized by sustained sinusoidal oscillations with a frequency around 150 Hz, and a pulse song consisting of trains of short pulses with two distinct shapes, produced at a regular interval of around 40 ms[36]. Substrate-borne vibrations are associated

[1]ENI-G, a Joint Initiative of the University Medical Center Göttingen and the Max Planck Institute for Multidisciplinary Sciences, Göttingen, Germany. [2]Department of Neuroscience, Faculty VI, University of Oldenburg, Oldenburg, Germany. [3]These authors contributed equally: Elsa Steinfath, Afshin Khalili. ✉e-mail: jan.clemens@uol.de

with abdominal quivering and are pulsatile like the pulse song, but with a longer interval of 150–200 ms[35]. Both song and vibration influence female mating behaviors and can therefore be considered signals[37]. In receptive females, song elicits acceptance behaviors such as slowing and vaginal plate opening[24,38]. Conversely, unreceptive females display rejection behaviors in response to song, including acceleration and ovipositor extrusion[27,39]. Although the evidence for vibrations is less conclusive and requires further investigation, several studies indicate that vibrations also elicit female acceptance behaviors, including slowing and copulation[35,40], suggesting that vibration similarly functions as a signal.

Despite evidence that both signals affect female behavior, how the male brain coordinates air-borne song and substrate-borne vibration remains unknown. In the *Drosophila* brain, sexual behaviors are controlled by sexually dimorphic neurons that express the transcription factors *fruitless* or *doublesex*[35,41–44]. The neural circuitry underlying courtship song production is well understood, with central neuron types P1a and pC2l integrating social cues—chemical, visual, acoustic—to drive persistent courtship and singing[24,31,33,45,46] in the ventral nerve chord (VNC) via at least two descending neurons (DNs), pIP10[23] and pMP2[47]. The choice between the two song modes is driven by the relative activity of these DNs and by circuit dynamics in the VNC[33,48].

In contrast, the behavioral contexts and neural circuits that drive vibration in *Drosophila* males are unknown. It is unclear to what extent song and vibration are produced simultaneously or sequentially since recordings of both signals with sufficient temporal resolution in naturally interacting animals are lacking. Because vibrations are associated with abdominal quivering rather than wing movements like the song[35,49] they are likely generated by a separate motor program.

## Results

### Simultaneous recordings of song and vibration during courtship in *Drosophila*

To assess the coordination of song and vibration, we designed a behavioral chamber that can reliably record song and vibration simultaneously (Fig. 1A–C, S1C, modified from[27,50]). Microphones tiling the behavioral setup floor were covered by a thin paper serving as a substrate for the flies to walk on and for transmitting both signal types. We discriminated song and vibration pulses based on their interval differences, whereby song pulses arrive at intervals between 30 and 45 ms, and inter-vibration intervals (IVIs) are much longer and range between 140 and 180 ms (Fig. 1D). Using laser vibrometry, we observed IVIs matching previous readouts of vibrations[35], (Fig. S1A, B). By recording high-resolution video of courtship in a smaller chamber and analyzing the movement of the abdomen during vibrations using SLEAP pose tracking[51], (Fig. S1D, E) we confirmed that the vibration pulses are associated with the previously reported abdominal quivering[35].

### Male flies dynamically switch between song and vibration during courtship

With access to song and vibration produced by the male during courtship, we next characterized the coordination between these two signals. During courtship, males vibrated twice as much compared to singing, and the vibration bouts were longer than song bouts (Fig. 1F, G). Song is produced using uni-lateral wing extensions, while vibrations do not require the wings (Fig. 1E, S1G, H). Although 19% of vibrations occurred while the wing was extended, males rarely sang and vibrated at the same time (1%) (Fig. 1H, S1F), indicating that the male is physically able to simultaneously sing and vibrate but chooses not to overlap both signals.

The male switched dynamically and non-randomly between sine, pulse and vibrations (Fig. 1I). Transitions between the song modes (sine, pulse) were more frequent (26% of all transitions) than transitions between song and vibration (only 7% of all transitions). Moreover, while pulse and sine were sequenced into bouts with no or very short pauses, vibrations were separated from song by a pause of around 1 s (Fig. S2). This temporal coordination of song and vibration suggests that these two signals are produced in distinct behavioral contexts. To identify these contexts, we next linked recordings of song and vibration with video tracking of the courtship interactions using computational modeling.

### Locomotion and distance of the female fly determine signal choice

The choice between sine and pulse song is based on male locomotor state and female behavior[27,36,52] and our analyses of the transitions between song and vibration suggest that this might also be true for vibrations (Fig. 1I). To identify the cues that inform the male's choice between song and vibration, we employed generalized linear models (GLMs) using the dynamics of social cues extracted from the male and female tracking data to predict the male's choice between song, vibration, or no signal (Fig. 2A, B).

A model fitted using all 19 cues predicted the male's choice to vibrate with only few confusions (83% correct), demonstrating that vibrations, just like song, are produced not randomly but in a context-dependent manner (Fig. 2C). Most errors were false positives (predicted song or vibration during "no signal"), implying that additional factors, such as stochasticity or internal states, further contribute to the male's signal choice[52]. To assess the contribution of individual cues to the signal choice, we fitted individual models for each cue (Fig. 2D–F, S3A, B) and found that models fitted with male or female locomotor cues predicted vibrations best, with 83-92% accuracy, while relative cues like distance and orientation were less predictive (<50%). In contrast, song was predicted best by the relative cues distance and orientation (71%), less well by male cues (38–49%), and poorly by female cues (12–18%). These findings indicate that male and female locomotion, rather than their distance or angle, are the strongest determinants of vibrations.

We then determined how the cues influence signal choice by examining the integral of each cue's filter. If the sign of the integral is positive, then high cue values (e.g., large distances) promote the signal; if the sign is negative, then the cue suppresses the signal. The filters for the best male and female predictors—female velocity and male lateral velocity—were positive for song and no signal but negative for vibrations (Fig. 2G, H). This trend was consistent for all locomotion filters (Fig. S3B), indicating that males tend to vibrate when they or the female are slow or stationary, and they tend to sing when either the male or female is moving (Fig. 2I, S3B–D). The observed association between stationarity and male vibration production is not due to limitations in our recording setup (Fig. S1E) and is consistent with previous findings linking female immobility to increased male vibration behavior[35].

The filter for distance, the cue most predictive of singing, was negative for singing and positive for vibrations, indicating that males vibrate when farther away from the female and sing when in closer proximity (Fig. 2G–I, S3C–F). In addition, the distance filter for song changed its sign from positive to negative, indicating that a reduction in distance to the female drives singing (Fig. 2H). This is consistent with singing frequently preceding copulation attempts, during which a previously stationary male moves closer to the female[39]. Distance is known to determine the choice between song types[27,36], as well as the amplitude of song[53]. It also determined the choice between song and vibration, indicating its centrality for courtship signal choice. Interestingly, the context in which males vibrate—slow and far from the female—was previously interpreted as a disengaged state[52]. Having access to vibrations during courtship, we found that part of this 'passive' state is not idle, but that the male actively signals to the female.

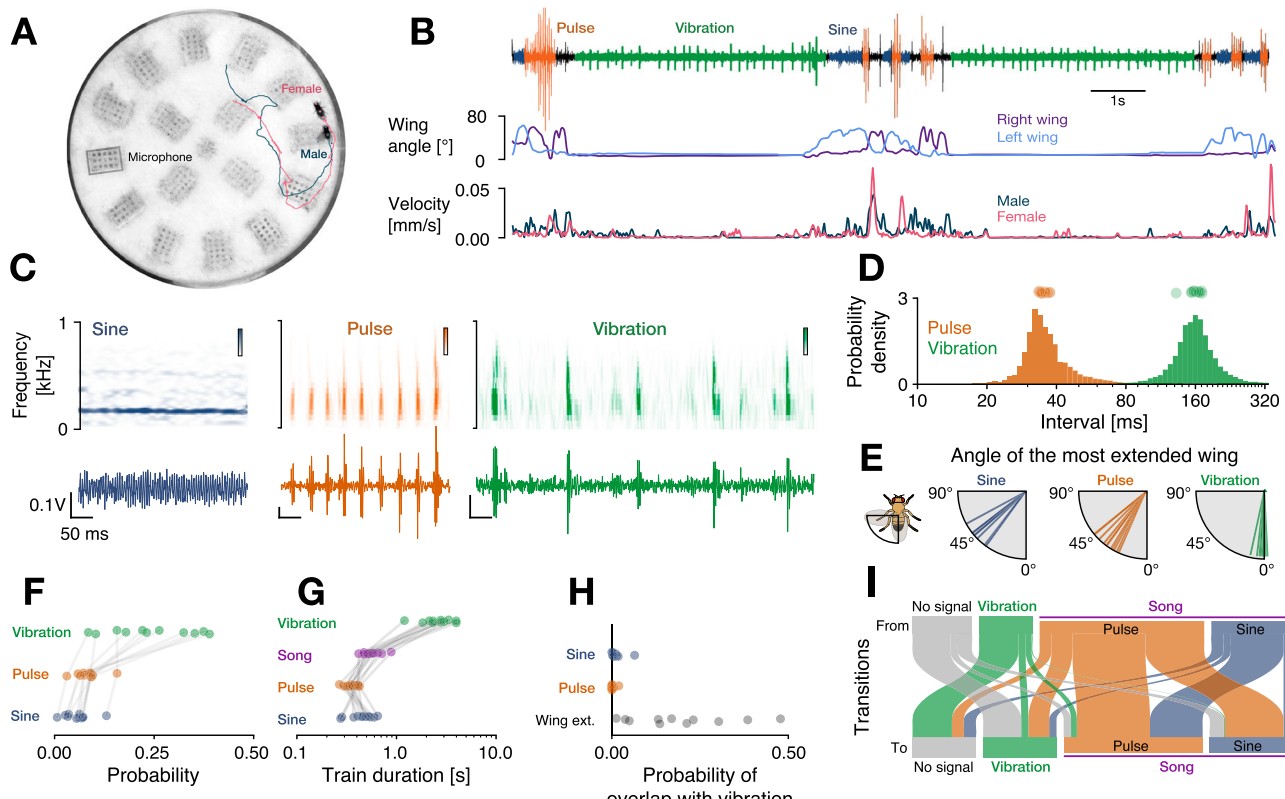

**Fig. 1 | Drosophila males produce two multimodal signals—song and vibration—during courtship.** **A** Behavioral chamber with a male (blue) courting a female fly (pink), tracked poses (dots) and walking trajectories (lines). Gray box marks one of 16 microphones embedded in the floor. **B** Audio trace (top) from one microphone with sine song (blue), pulse song (orange), and vibrations (green) alongside behavioral cues extracted from pose tracking: male's left and right wing angle (middle) as well as male and female velocity (bottom). **C** Waveforms (bottom) and spectrograms (top) for sine song (blue), pulse song (orange), and vibrations (green). **D** Distribution of intervals between song pulses (orange, $N = 27310$) and vibrations (green, $N = 16785$). Dots on top show median values for each male. Intervals between song pulses ($35.5 \pm 11.4$ ms, median ± interquartile range (IQR)) are much shorter than intervals between vibrations ($160 \pm 41$ ms). **E** Median angle of the most extended wing during sine ($58 \pm 9°$, median ± standard deviation), pulse ($48 \pm 9°$) and vibration ($12 \pm 4°$). Values close to 0° correspond to no wing extension. Males rarely extend their wings during vibrations. **F** Probability of producing sine ($6 \pm 3\%$, mean ± standard deviation), pulse ($8 \pm 3\%$), vibration ($24 \pm 10\%$), and no signal ($62 \pm 11\%$) during courtship. Males produce more vibrations than song (sine+pulse, $p = 0.02$). **G** Duration of sine songs ($460 \pm 145$ ms), pulse trains ($355 \pm 79$ ms), song bouts ($562 \pm 129$ ms), and vibration trains ($2785 \pm 944$). Vibration trains are longer than song bouts ($p = 5 \cdot 10^{-4}$) **H** Overlap between vibrations and sine song ($0.012 \pm 0.017$), pulse song ($0.002 \pm 0.006$) or wing extensions ($0.19 \pm 0.14$). **I** Transitions between no signals (gray), vibration (green), pulse (orange), and sine (blue). Line width is proportional to the probability of transitioning from one signal (top) to another (bottom). Transitions between the song modes (pulse and sine) are more frequent than between song and vibrations ($p = 5 \, 10^{-4}$). $N = 11$ males in D–I. All reported p-values from one-sided Wilcoxon tests. Reported summary statistics correspond to mean ± standard deviation (std.) unless noted otherwise.

## Stationarity is necessary and sufficient to drive vibrations in males

The statistical models of male signal choice showed that stationarity predicts vibrations (Fig. 2). However, it is possible that other behaviors that females primarily perform when stationary (e.g., grooming) could be the cause of vibrations. We therefore causally tested the role of stationarity by manipulating locomotion during courtship. According to the behavioral models, stopping the male or the female should increase the probability of observing vibrations, while inducing locomotion should suppress vibrations (Fig. 2G–I). To not interfere with the male's signaling ability, we optogenetically manipulated female walking behavior during courtship.

We first stopped the female by expressing GtACR1, an inhibitory channelrhodopsin, in all motor neurons using the vGlut driver[54]. Stopping the female increased vibrations by 30% (Fig. 3A, B). Conversely, inducing female walking by optogenetically activating the DNp28 neurons[55,56] nearly abolished vibrations (Fig. 3C, D). These causal interventions therefore confirmed that stationarity is necessary and sufficient for vibrations. Further, singing was best predicted by male-female distance (Fig. 2F), but distance changed only little

when stopping the female (Fig. S4B). Distance did increase when inducing female locomotion, and this weakly suppressed singing (Fig. S4C), demonstrating that controlling female locomotion only weakly affected singing behavior (Fig. S4A–C), consistent with the behavioral models (Fig. 2E, F). In summary, locomotion controls vibrations.

Although we genetically controlled female locomotion, the male chases the female and his movement is tightly correlated to her movement (Fig. 3A, C), in short, stopping the female during courtship also stops the male. This correlation also explains why both male and female locomotor cues predict vibrations (Fig. 2F, S3A). However, male signal choice is more strongly determined by his own than by the female's stationarity (Fig. 2D, S4D): Male velocity distributions are clearly distinct when he sings versus vibrates, while female velocity distributions overlap considerably during song or vibration. It is therefore likely that the male's locomotor state controls the choice between song and vibration, and is not influenced by the female movement. This co-regulation of locomotion and signaling likely evolved because walking can interfere with the transmission and perception of vibrations[40].

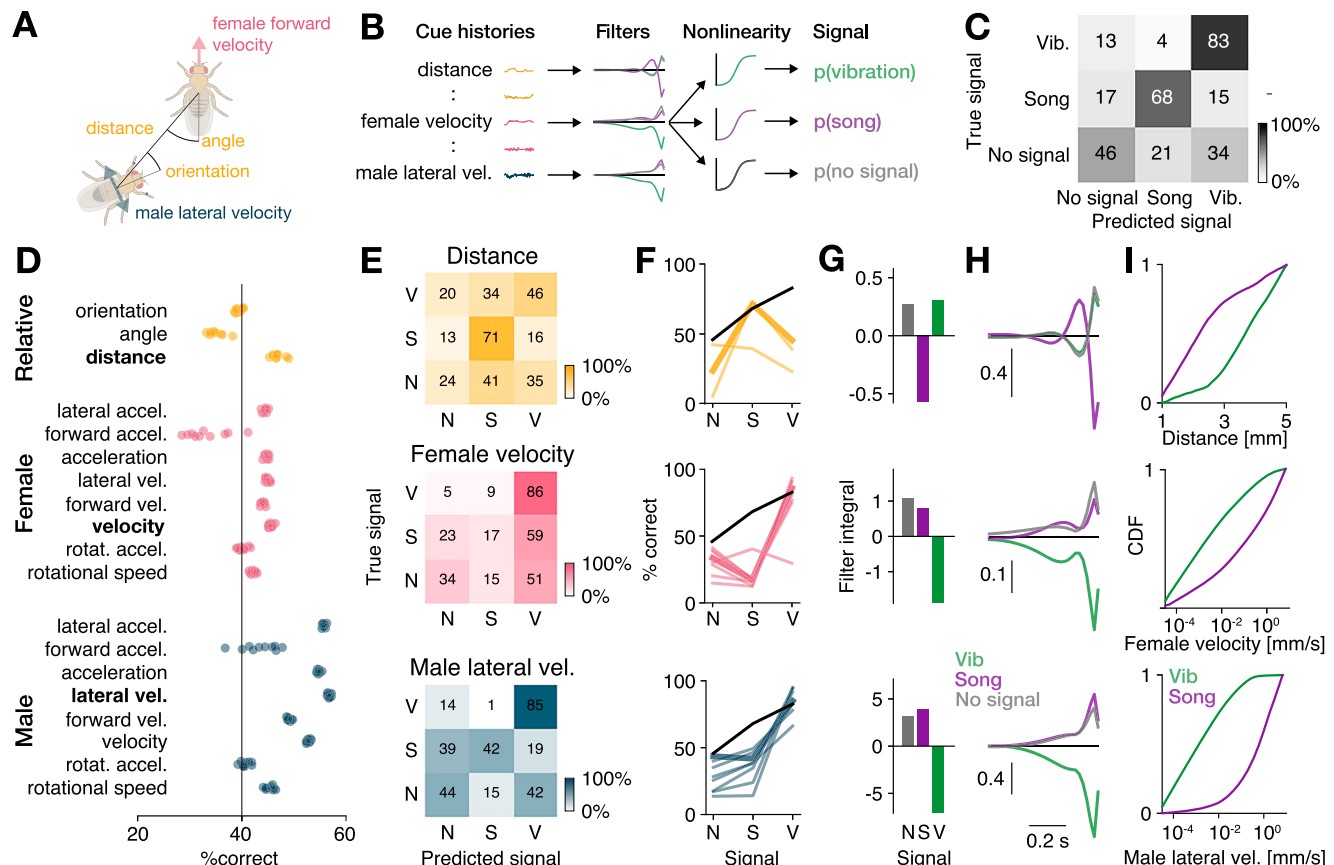

**Fig. 2 | Locomotion and distance predict signal choice. A** Examples of feedback cues used to predict the male's signal choice. **B** Signal choice (vibration, song, no signal) was predicted using the cues histories (**A**) from one second preceding each time point. Choice relevant temporal cue patterns were detected using filters, with one filter per cue and signal type. The filtered cues are then passed through a nonlinearity that yields the probability of observing each signal. **C** Confusion matrix for a model fitted to predict the male's signal choice from all cues. Shading and numbers indicate the classification percentage (see color bar). **D** Accuracy (% correct) for individual male (blue), female (pink), and relative (yellow) cues. Models were fitted to predict male signal choice using individual cues only. Dots correspond to result from 10 model fits from independent train-test splits. **E** Confusion matrices for predicting the male's signal choice (V - vibration, S - song, N - no signal) using the most predictive individual male cue (lateral velocity, bottom), female cue (female velocity, middle), and relative cue (distance, top). Shading and numbers indicate the classification percentage (see color bar). **F** Signal-wise performance for male (bottom), female (middle), and relative (top) cues. Male cues predict vibrations very well and song moderately. Female cues only predict vibrations well, and relative distance predicts song well. Thick colored lines correspond to the best cue for each cue group shown in (**E**). Black lines show the performance of the multi-feature model from (**C**). See also Fig. S3A. **G** Integral over the filters for each signal for the cues shown in (**E**). Small male (bottom) and female velocity (middle) values predict vibration. Small male-female distances (top) predict song. **H** Filter shapes of the cues shown in (**E**). The distance filter for song changes its sign from positive to negative, indicating that a reduction in distance drives song. **I** Cumulative density functions (CDFs) for the cues shown in (**E**). Vibrations are produced at low velocities (bottom, middle), and song is produced at smaller distances (top).

## Central "song" neurons drive song and vibration with complex dynamics

Having shown that locomotion regulates the switch to and from vibration, we next asked how this switch is implemented in the fly brain. While the neurons in the central brain that drive singing have been identified[23,33], cell types that drive vibration are unknown. To test whether song and vibration are driven by distinct or overlapping central circuits, we examined whether key neurons of the song pathway also drive vibrations.

Several cell types that express the sex-determination genes *doublesex* and *fruitless*[41–43,57–59] integrate social cues and drive singing in males. We focused on two brain-local neurons and two descending neurons that drive singing when activated. The pC2l neurons in the central brain process auditory and visual cues and elicit robust singing via a direct connection to the descending pIP10 neurons[24,33,36,44,47,60,61]. the P1a neurons[23,33,36,46,61] process pheromones[45,62,63] and likely receive input from pC2l neurons[33]. P1a neurons induce a persistent arousal state that can drive courtship and singing, or aggression[64,65], on two timescales: on the order of up to ten seconds, via slowly decaying activity in P1a itself[63] and on the order of up to a minute via a recurrent neural network downstream of P1a[31]. P1a neuron activation alone tends to yield only a little song since it drives song indirectly, via a disinhibitory circuit motif[31,33,36,64]. The decision to sing, encoded in the activity of pC2l and P1a neurons, is relayed to premotor circuits in the VNC via at least two descending neurons: pIP10 and pMP2[23,60]. The pIP10 neurons receive inputs from the pC2l neurons, but the central inputs to pMP2 or downstream targets of P1a are unknown.

Activation of all *doublesex* and *fruitless* neurons induces vibrations[35], but specific cell types—and hence circuits—that drive vibrations were not known. We optogenetically activated P1a[64], pC2l[44], pIP10[23], and pMP2[47] in solitary males with varying light intensities and examined the time spent producing each of the communication signals—vibrations, pulse, sine—during and between activations (Fig. 4B). The activation of the descending neurons pIP10 or pMP2 drove song but no vibrations. However, the two central brain neurons, P1a and pC2l, elicited both song and vibration. Among males with activated pC2l neurons, 8 out of 25 vibrated, and all 35 males with activated P1a neurons vibrated. This suggests that multimodal signal generation is

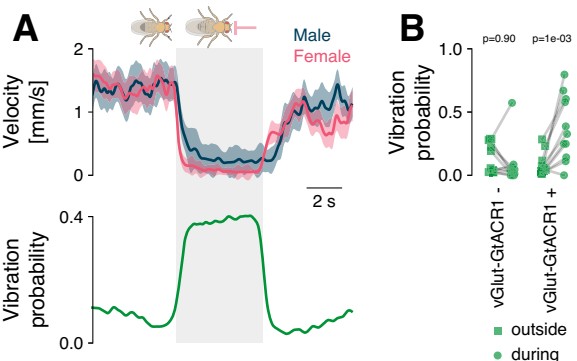

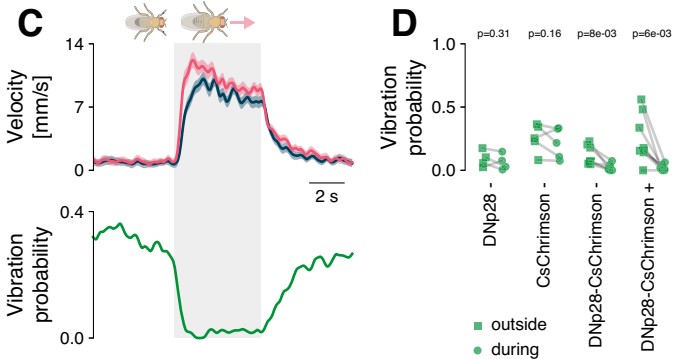

**Fig. 3 | Immobility is a necessary and sufficient trigger of male vibrations.**
**A** Optogenetic inactivation (gray) of all motor neurons (MNs) in a female courted by a wild-type male stops the pair (top, male/female velocity blue/pink) and triggers male vibrations (bottom). Females expressed GtACR1 in all glutamatergic neurons. Optogenetic stimulus 525 nm at 14 mW/cm². **B** Average vibration probability outside of (squares) and during (circles) optogenetic inactivation of the MNs. Control females (vGlut-GtACR1-) had the same genotype but were not fed all-trans retinal, a cofactor required to make GtACR1 light sensitive. Lines connect data from the same pair during the different epochs (vGlut-GtACR1 atr- $N = 11$, atr+ $N = 11$). $P$-values from a paired, one-sided Wilcoxon test of the hypothesis that the vibration probability increases due to female slowing. **C** Optogenetic activation (gray) of DNp28 neurons

in a female courted by a wild-type male accelerates the pair (top, male/female velocity blue/pink) and suppresses male vibrations (bottom). Optogenetic stimulus 625 nm at 89 mW/cm². **D** Average vibration probability outside of (squares) and during (circles) optogenetic activation of DNp28. Lines connect data from the same pair during the different epochs (DNP28-Gal4+ $N = 4$, UAS-Chrimson+ $N = 5$, DNp28-Chrimson- $N = 7$, DNp28-Chrimson+ $N = 9$). $P$-values from a paired, one-sided Wilcoxon test of the hypothesis that the vibration probability decreases due to female acceleration. Lines and shaded areas in (**A**) and (**C**) show the mean ± standard error of the mean. A '+'/'-' after the genotype names in (**B**) and (**D**) indicates the presence/absence of all-trans retinal in the food.

orchestrated by a shared neural circuit capable of driving both signals. Consequently, descending neurons engage distinct motor circuits in the ventral nerve cord, dedicated to either song production or vibration.

We next examined the dynamics with which P1a and pC2l drove multimodal signals. Activating P1a neurons[64] reliably induced vibrations that outlasted the activation for tens of seconds (Fig. 4C, D, S5A, B), independent of activation strength (Fig. S5E). Our sparse activation protocol also resulted in a few song bouts during and after activation. This implies that the persistent courtship state induced by P1a neuron activation jointly controls the multimodal courtship signals of song and vibration[31,64]. By contrast, pC2l neuron activation reliably drove song (Fig. 4E, F, S5C, D). Interestingly, at the offset of strong activation, we observed vibrations lasting 5–10 s (Fig. 4F). The pC2l neurons are known to produce sine song at activation offset[24,33,47], but this sine song is much shorter (< 1 s) than the vibrations (5–10 s) (Fig. S5D). In addition, pulse-sine-vibration sequences were rare, and most transitions into vibrations were preceded by pulse song rather than sine song (Fig. S5G–J).

Thus, the previously identified "song circuit" comprised of P1a and pC2l neurons drove multimodal signals. The pC2l neurons directly drove song, P1a directly drove vibrations. The cell type-specific dynamics likely reflect differences in downstream connectivity. As pC2l neurons drive song via a direct connection to pIP10[33,66], we hypothesize that P1a neurons similarly drive vibrations via an unknown descending neuron (DNvib). Further, pC2l drives offset sine via its connection to P1a neurons, disinhibiting ventral nerve chord sine nodes[33]. We hypothesize that the offset vibrations are also driven through this pC2l-P1a connection and the DNvib.

**Central P1a neurons jointly control male locomotion and vibration**
Signaling needs to be coordinated with ongoing behaviors to ensure its efficacy, e.g, vocalizations are coordinated with breathing in vertebrates[29,67]. Our behavioral analyses (Figs. 2, 3) showed that stationarity triggers vibrations, and P1a neuron activation is known to induce locomotor arrest in males[36,64]. This suggests that P1a neurons not only drive multimodal signals but also coordinate them with locomotion. This could be attributed to P1a neurons either controlling

locomotor state and vibrations in parallel or inducing a vibration motor program that inherently includes stopping the male (Fig. 4G). In the first case, P1a neuron activation should stop males, but not all stationary males should vibrate. In the other case, all males that stop upon P1a neuron activation should also vibrate. We therefore examined the association between P1a neuron activation, male locomotion, and vibrations. We find that P1a neuron activation induced locomotor arrest in solitary males[36] in almost all males (Fig. 4H–J, S6). However, only ~60% of the stationary males vibrated independent of activation strength (Fig. 4J), suggesting that P1a neurons do not induce a drive to vibrate, which in turn stops males. Instead, P1a neuron activation induces two distinct motor programs: one that near-deterministically stops the male and puts him into "vibration mode" and another that then probabilistically triggers vibrations within this state. However, this does not rule out the possibility that the locomotor state itself inhibits vibrations through an additional gating mechanism in walking males. Activation of pC2l neurons does not strongly affect locomotion, but males stop at activation offset, likely because pC2l neurons drive vibrations through P1a neurons (Fig. S5K). Thus, P1a neurons coordinate signaling with ongoing behavior—they stop males and induce vibrations.

**Mutual inhibition coordinates song and vibration**
During natural courtship and during optogenetic activation, song and vibration rarely overlap (Fig. 1H), raising the question of how the song and vibration pathways interact downstream of P1a and pC2l neurons. A common circuit motif that prevents the simultaneous expression of two behaviors is mutual inhibition[68,69] and might be at work downstream of P1a and pC2l. More specifically, we predicted that P1a neuron activation would suppress song since it drives vibrations, and pC2l neuron activation would suppress vibrations, given that it drives song (Fig. 5A, B). To unmask mutual inhibition between the song and vibration pathways, we activated P1a and pC2l neurons not in solitary males but in males paired with a female. We hypothesized that the presence of the female would drive P1a and pC2l neurons, consequently triggering courtship with song and vibration (Fig. 5C, D, S7A, B). Consistent with our prediction, P1a neuron activation strongly suppressed song (Fig. 5C, E) by interrupting singing in all flies, even in those that did not switch to

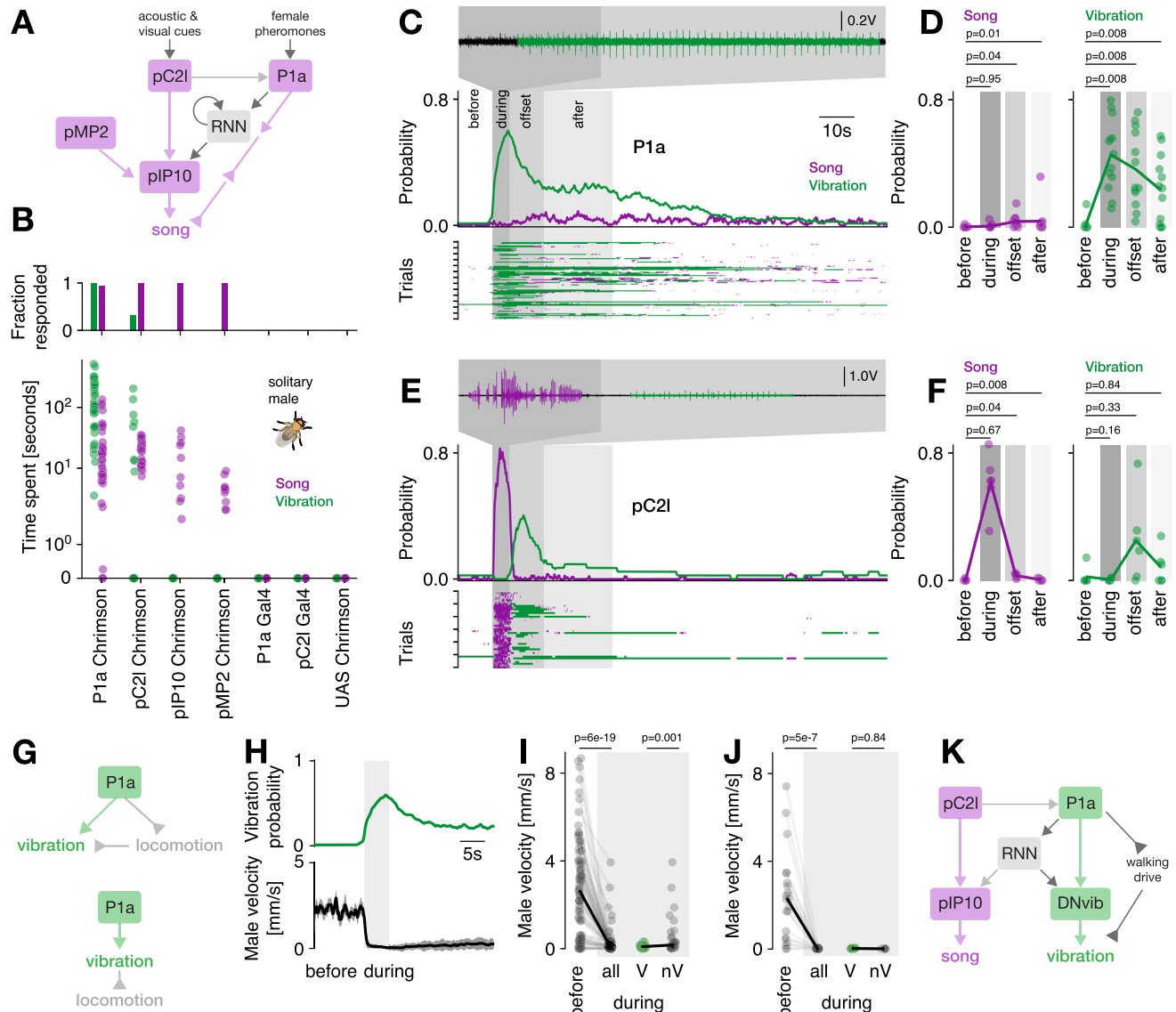

**Fig. 4 | Dynamical multimodal signaling is controlled by a network including P1a and pC2l neurons. A** *Drosophila* song circuit: pC2l drives song via pIP10; pMP2 also drives song. P1a drives song indirectly, via a recurrent neural network (RNN) and a disinhibitory motif. Arrows indicate excitatory (regular) or inhibitory (inverted) connections. **B** Song (purple) and vibration (green) evoked by optogenetic activation of P1a, pC2l, pIP10 and pMP2 across different light intensities. Top: fraction of males singing or vibrating. Bottom: average time individual males spent producing each signal. Y-axis symlog scaled. N=35/25/10/10/6/5/5 flies P1a/pC2l/pIP10/pMP2-Chrimson, three controls. **C** Microphone recording (top), trial average probability (middle), and single trial raster (bottom) showing song (purple) and vibration (green) evoked by P1a activation in solitary males (27 mW/cm², N = 13 flies, 7 trials/fly). Gray shadings delimit analysis epochs in (**D**). **D** Song (left) and vibration (right) probability across epochs relative to P1a activation onset: before -10–0, during 0–5, offset 5–15, after 15–35 s, same data as (**C**). **E, F** Same as (**C, D**) but for

pC2l (83 mW/cm², N = 6 flies, 7 trials/fly). **G** Two hypotheses regarding vibration and locomotion control. Top: P1a independently controls vibration and suppresses locomotion. Bottom: P1a drives a single motor program, causing both stopping and vibration. **H** Vibration probability (top) and average male velocity (bottom) in response to P1a activation (same data as C). **I** Male velocity before and during P1a activation. Dots correspond to trials. Right: split into vibrating (green, V) and non-vibrating males (black, nV) (same data as (C)). **J** Same as I but for stronger P1a activation (209 mW/cm², N = 3 flies). **K** Working model of multimodal signaling: P1a drives vibrations directly and persistently, through direct and indirect (via RNN) connections with an unidentified descending neuron DNvib. Additionally, P1a controls locomotion to tie vibrations to male stationarity. *P*-values in (**D, F**) from one-sided Wilcoxon test testing whether song or vibration probability increases; in (**I, J**) from one-sided Mann-Whitney U tests of the hypothesis that P1a activation slows males, and that vibrating males are slower.

vibrations (Fig. S7C). Conversely, pC2l neuron activation almost completely suppressed vibrations (Fig. 5D, F). Almost all flies that were vibrating in the five seconds prior to activation ceased their vibrations, even if they did not initiate singing behavior (Fig. S7D). These results show that mutual inhibition reduces the overlap between multimodal signals in *Drosophila*. Mutual inhibition also coordinates the rapid switching between pulse and sine song[33], suggesting that similar circuit principles coordinate signal production across timescales.

## Circuit dynamics bias signaling and can be overridden by female cues for context-appropriate signaling

Optogenetic activation engaged a circuit with strong autonomous dynamics (Fig. 4C–F): P1a neurons drive vibrations during and for tens of seconds after activation and only a little song in solitary males. pC2l neurons drive a sequence of song during, and vibrations for 5–10 s after activation. However, signal dynamics during natural courtship with a female are much more variable (Fig. 1). For instance, the pulse to vibration transitions produced by pC2l activation (Fig. 4E) are rarely

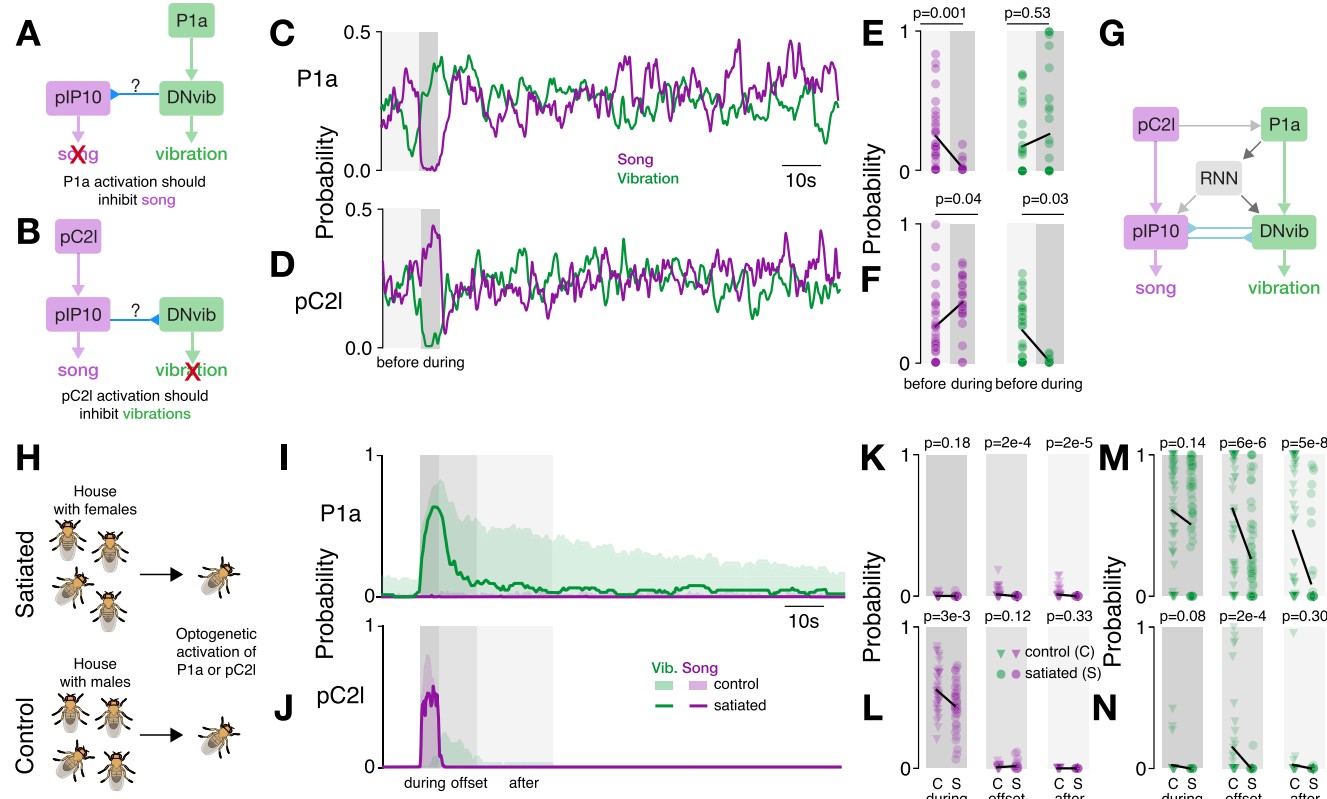

**Fig. 5 | Coordination and modulation of song and vibration via mutual inhibition, female cues, and motivational state. A, B** Hypothesized effects of mutual inhibition. Activation of P1a drives vibration and inhibits song (**A**). Activation of pC2l drives song and inhibits vibrations (**B**). Mutual inhibition is depicted as acting directly between descending neurons, but could also act downstream. **C, D** Probability of song (purple) and vibration (green) in courting males during optogenetic activation of P1a (**C**) or pC2l (**D**). Shaded areas indicate the epochs used for statics in (**E**) and (**F**). Only probabilities during which the male courted were included. Light intensity 27 mW/cm² at 625 nm. **E, F** Comparison of song (left) and vibration (right) before (10 s) and during (5 s) activation of P1a (**E**) and pC2l (**F**) in courting males.

The statistical tests only included trials in which the males courted the female before and during activation. *P*-values from a two-sided Wilcoxon test. **G** Working model of multimodal signaling with mutual inhibition. **H** Males were sexually satiated by housing them with 10–15 virgins 4–6 h prior to the experiments. Control males were housed with 10–15 males. **I, J** Probability of observing song (purple) and vibration (green) in sexually satiated (lines) and naive (shaded areas) solitary males upon optogenetic activation of P1a (**I**) and pC2l (**J**). Gray shaded areas indicate epochs used for statics in (**K–N**). **K, L** Comparison of song evoked in different time windows for P1a (**K**) and pC2l (**L**) in sexually satiated (S) and control (C) males. **M, N** Same as (**K, L**), but for vibrations.

seen during natural courtship (Fig. 1I). While we cannot rule out that consistent optogenetic activation of P1a and pC2l drives the circuit into a non-typical dynamical regime[70,71], we believe the differences in signal dynamics between optogenetic activation and natural courtship arise because P1a and pC2l are activated by intermittent and dynamical social cues from the female. Specifically, P1a responds to contact and volatile pheromones[45,62,63], while pC2l responds to acoustic and visual cues[24,33,61]. To assess how dynamical social cues modulate the circuit's autonomous dynamics during courtship, we assessed the data from activated P1a and pC2l neurons in males that courted a female (Fig. S7C, D). In the courting males, we found that activation of P1a or pC2l neurons did bias subsequent signaling towards vibrations. However, the bias was relatively weak and not as persistent as in solitary males (compare Fig. 4C, E). Thus, the circuit driving song and vibration in the central brain enables persistent yet flexible signaling. In the absence of social cues, activation of the circuit drives autonomous dynamics that enable persistent signaling. However, external cues can override these circuit dynamics to enable context-appropriate dynamical signaling.

## Song and vibration are under common motivational control
The persistence of courtship in *Drosophila* is driven by P1a neurons and modulated by sexual satiation, which reduces the initiation and persistence of courtship in males[72]. The effect of satiation is mediated by dopamine and leads to a reduced excitability of P1a neurons[63,72] as well as less persistence in P1a neuron activity itself[63] and in the recurrent

circuitry downstream of P1a neurons[31,73]. One advantage of driving song and vibration through a shared circuit is that only a few circuit nodes need to be manipulated to globally up- or down-regulate multimodal signaling. However, direct effects of sexual satiation on singing and vibration have not been investigated. To assess whether motivational state modulates the persistence of both signals, we induced sexual satiation by allowing males to freely mate with females, and subsequently activated P1a and pC2l neurons (Fig. 5H). We found that sexual satiation strongly reduced the persistence of both song and vibration (Fig. 5I–N). Satiated males were less likely to vibrate after P1a neuron activation, and their tendency to sing was even further diminished (Fig. 5I, K, M). For pC2l activation, satiation weakly reduced the singing and almost completely abolished vibrations after activation offset (Fig. 5J, L, N). An effect of sexual motivation on P1a neurons has been demonstrated previously[63,72,73] and we now show that satiation globally reduces the persistence of signaling during courtship, implying that both song and vibration are under common motivational control.

## A neural circuit model for multimodal signaling
Our experiments revealed a neural circuit that drives multimodal signals with complex and persistent dynamics. To test whether this circuit is indeed sufficient to explain the dynamics of multimodal signaling in *Drosophila*, we implemented a proof-of-concept circuit model (Fig. 6A, S8). The proposed model consisted of three major components: First,

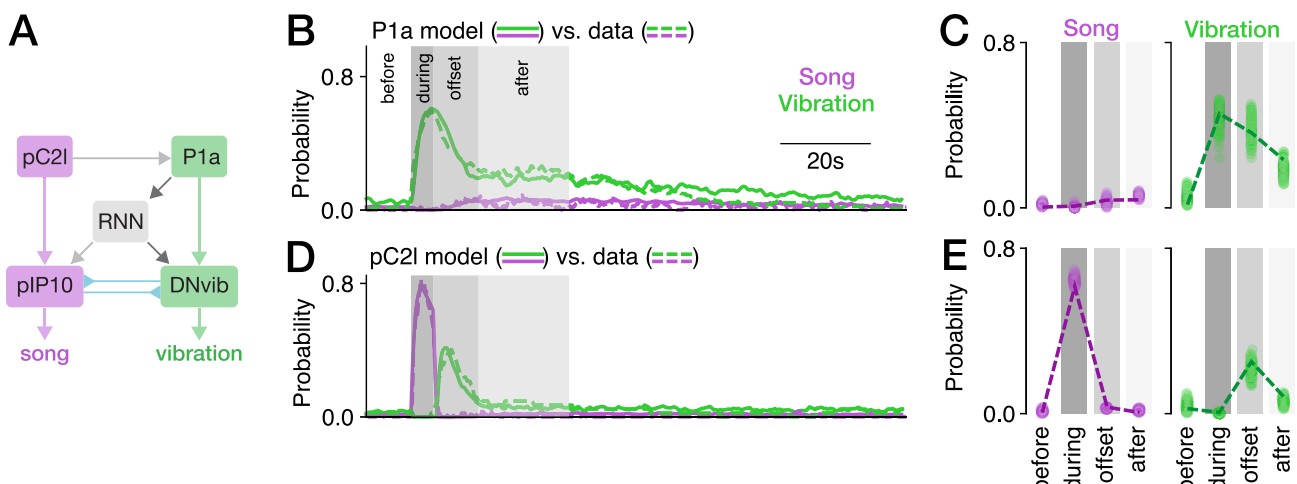

**Fig. 6 | A neural circuit model proposes elementary computations underlying multimodal signaling. A** Network diagram of the circuit model. Regular and inverted arrow heads indicate excitatory and inhibitory connections, respectively. **B, D** Song (purple) and vibration (green) for activation of P1a (**B**) and pC2l (**D**) in the model (solid lines) and the data (dashed lines, data from Fig. 4C, E). The model reproduces the data well: The mean-squared error between model and data is 0.003 for all traces. **C, E** Probability of observing song (purple) and vibration (green) in different epochs around the activation of P1a (**C**) and pC2l (**E**) in the model (dots correspond to model runs with independent noise) and the data (dashed lines, data from Fig. 4D, F).

at the top of the hierarchy are pC2l and P1a neurons, which are activated by social cues (or optogenetically) and drive song and vibration (Fig. 4C, E). Direct connections between pC2l and P1a neurons and descending neurons mediated the immediate effects of social cues or optogenetic activation in our experiments. pC2l is directly connected to pIP10, which drives song in the VNC[66]. Given that P1a neurons strongly drove vibrations with little delay (Fig. 4C), we hypothesized that P1a neurons are connected to an unknown vibration descending neuron, that we called DNvib. Second, all indirect effects of optogenetic activation—the vibrations at the offset of pC2l neuron activation, as well as the persistent song and vibration after P1a activation—were mediated by P1a neurons in our model. P1a neurons are known to drive slow circuit dynamics in two ways: Intrinsically, through the slow decay of P1a neuron activity itself, which lasts 5–10 s[63]. And extrinsically, through a recurrent neural network (RNN) downstream of P1a neurons that maintains activity for several tens of seconds[31,64]. The timescale of the intrinsic decay matched the timescale of offset vibrations after pC2l neuron activation. Behavioral[33] and female connectome data (Fig. S9)[74,75] suggest that pC2l neurons likely weakly connect to P1a. In our model, pC2l activation induces slowly decaying activity in P1a neurons, but is unable to engage the RNN downstream of P1a neurons. We implemented this by requiring the RNN to receive strong P1a activation for persistent activity. The RNN in turn mediates the persistent production of multimodal signals through connections to descending neurons that control song and vibration. Lastly, the inhibitory cross-talk between song and vibration was mediated by mutual inhibition downstream of pC2l and P1a neurons (Fig. 5C–G), likely at the level of the descending pathways or in the premotor centers in the VNC[47]. In the model, we implemented this as mutual inhibition between pIP10 and DNvib neurons. Activation of pC2l neurons activates pIP10 neurons, and pIP10 neurons drive song. but also inhibit DNvib neurons and hence vibrations. Conversely, activation of P1a neurons activates DNvib neurons, which drive vibrations and inhibit pIP10 neurons and thereby song. Adaptation and noise in the mutual inhibition can enable bistable dynamics[69], which in our model leads to switching between song and vibration.

This model successfully reproduced the behavioral dynamics. Activating the model P1a neurons produced vibration, followed by a persistent phase of mainly vibration and only a little song, which both decay over time (Fig. 6B, C). Activation of pC2l neurons in the model yielded song, directly followed by vibrations (Fig. 6D, E, S10A–C). The persistent phase was mediated by the RNN (Fig. S8). Ablation of the RNN nearly abolished signals after P1a neuron activation during the persistent phase, but did not strongly affect the offset vibrations evoked by pC2l neuron activation and mediated via the slowly decaying dynamics intrinsic to P1a neurons (Fig. S10D–F). Mutual inhibition was required in the model to reduce the overlap between song and vibration, as in natural courtship (Fig. S2, 5C–F), and in the model, vibrations were suppressed when pC2l neurons were activated and song was suppressed when P1a neurons were activated (Fig. S10G–I). The circuit model also reproduced motivational effects in the circuit (Fig. 5I–N). Reducing the excitability of pC2l neurons, P1a neurons, and the recurrent network reduced song during pC2l neuron activation and strongly reduced the vibrations after activation of pC2l or P1a neurons (Fig. S10J–L). This neural circuit model replicated our behavioral findings and therefore provides insights into the circuit mechanisms that coordinate multimodal signaling behaviors.

## Discussion

We have identified the behavioral contexts (Figs. 2, 3) and circuit motifs that drive multimodal communication signals in *Drosophila* males (Figs. 4, 5, 6). This circuit generates signals with long-lasting, cell-type-specific dynamics (Figs. 4, 5), sets the locomotor state required for efficient signal transmission (Figs. 2G–I, 4G–K), and controls multimodal signals through motivational state (Fig. 5H–N).

We found that males produce vibrations when stationary (Figs. 2, 3), a context that previous studies interpreted as an idle state[52,64]. We show that males are not necessarily idle when sitting next to the female but actively produce communication signals, highlighting the importance of recording all behaviors for correctly interpreting behavioral contexts and the underlying neural circuits[36]. By vibrating primarily when he and the female are stationary and thus when the sender's and receiver's legs have full contact with the substrate, the male improves the transmission of vibrations: Vibrations are transmitted via the legs to the substrate, since the abdomen moves but does not touch/tap the substrate[35,76], and they are detected by leg mechanosensors in the female[40]. Walking, therefore, interferes with the transmission and detection of vibrations. Song, on the other hand, is airborne and its transmission is not impaired by walking (Fig. 2G–I). But since the song is detected using a highly directional sound receiver[77], it is produced at a more restricted set of positions (Fig. S3E, F). The

P1a neurons drive vibrations and induce male stationarity and therewith a locomotor state that favors the transmission of the vibrations (Fig. 4B–D, G–K). This coordination of signaling with ongoing behaviors like locomotion or respiration to optimize signal transmission is a general principle of behavioral control. For instance, vocalizations and respiration are coordinated in birds or mammals through shared circuits[29,30,78].

Female stationarity was previously[35,40,79] interpreted as the effect of vibrations, while our behavioral analyses (Fig. 2) and interventions (Fig. 3) show that it is the cause: Stopping the female during courtship is sufficient to drive male vibrations. Both findings can be reconciled: Song, often produced when the male chases the female, slows and stops her[24,80,81]. Vibrations, being produced when the female is stationary (Fig. 2)[35], might then prolong phases of stationarity. More experiments will be necessary to elucidate the behavioral effects of song and vibration and to identify the circuits that process both signals[79,82,83].

Multimodal signals are driven by an integrated neural circuit in *Drosophila*: The P1a and pC2l neurons—previously considered "song neurons"—drive song and vibration with complex and persistent dynamics (Fig. 4). Multimodal signaling via a single circuit is likely a general principle, since it facilitates signal coordination and modulation (Fig. 5). The periaqueductal gray (PAG) is hypothesized to control multimodal signaling in mammals and birds and shares properties with the proposed circuit in *Drosophila*[5]: The PAG drives vocalizations[22,84], integrates contextual and motivational information, and innervates multiple premotor regions that control different motor programs[5]. However, precise circuit interactions that might control multimodal signaling in the PAG remain to be identified.

We propose elemental motifs that coordinate multimodal signaling in *Drosophila* using genetic manipulations combined with a computational model. First, direct connections between P1a and pC2l, and descending neurons allow external sensory cues to directly and rapidly affect signaling (Figs. 2, 5A–F and S7). Visual motion cues from the walking female activate pC2l[33,61] to drive song when the male and/or the female move. Notably, song slows the female[24,81], thereby creating the behavioral context for vibrations. The song-vibration sequence evoked by optogenetic activation of pC2l (Fig. 4E) may therefore constitute a motor prior that facilitates this signal sequence. P1a activity is controlled via chemosensory inputs[45], but the specific cues that drive vibrations in P1a are unclear. The male is too far from the female for contact pheromones (Fig. S3E, F), but volatile pheromones reactivating P1a neurons in an aroused male might suffice[85].

Our experiments also showed that slow dynamics and recurrence act as a memory of the female cues and enable persistent courtship signaling in the absence of constant input from interaction partners (Fig. 4C–F,[86]). These motifs are also found in other systems and therefore likely constitute universal building blocks for controlling behavior: For instance, recurrent circuits in the ventromedial nucleus of the hypothalamus (VMHvl) of mice are central to generating persistent social behaviors that can be easily manipulated by sensory cues through line attractor dynamics[32,87,88]. While elucidating the precise circuit, cellular, and molecular mechanisms underlying these common dynamics is challenging in vertebrates models, it will be much more feasible in *Drosophila,* given that we have genetic access to identified cell types and connectomics[74].

Lastly, mutual inhibition downstream of P1a and pC2l—between the DNs (Figs. 5A–F, 6) or downstream in the VNC—coordinates multimodal signaling at the motor level to prevent the overlap between song and vibration (Fig. 1H). Notably, the same circuit motif coordinates song production and is likely implemented via pulse- and sine-specific TN1 neurons that inhibit each other in the VNC[33]. Mutual inhibition is a core motif whenever mutually exclusive behaviors or patterns of muscle activity are produced by the nervous system—during perceptual decision making, action selection, or motor pattern generation[68,69,89,90].

The descending pathways by which P1a controls locomotor state and vibrations remain to be identified. Unlike pulse and sine, which occur in complex bouts with rapid mode switches[33], direct/immediate transitions between song and vibration are rare during courtship (Fig. 1I, S2). Accordingly, neither pMP2 nor pIP10 drives vibrations (Fig. 4B), and vibrations are likely driven by an unknown DNvib (Fig. 6). The complete wiring diagrams of the male brain and VNC will facilitate the identification of descending pathways and pattern-generating circuits downstream of P1a that control multimodal signaling and locomotor state in *Drosophila*[74,91–93]. Ultimately, vibrations are likely produced by thoracic and abdominal contractions that are transmitted via the legs to the substrate[94]. The thoracic muscles, which include the wing muscles that are also required for singing[95,96], may therefore also contribute to vibrations[76] and may constitute, after the divergence of pathways at the premotor level, a convergent *final common pathway*[97] for multimodal signaling in *Drosophila*.

Overall, our results identify common circuit motifs—feedforward excitation, recurrence, mutual inhibition—that can be combined in a single circuit to support dynamical and context-specific multimodal signaling. Moreover, we establish *Drosophila* as a model system for studying multimodal communication.

## Methods
### Fly strains and rearing
Flies were kept on a 12:12 h dark:light cycle, at 25 °C and 60% humidity. Flies were collected as virgins within 8 h after eclosion, separated by sex, and then housed in groups of 3–15 flies. Fly genotypes are listed in Table 1.

### Behavioral setups
The behavioral chamber measured 44 mm in diameter and 1.9 mm in height; the chamber and lid were machined from transparent acrylic. Chamber lids were coated with Sigmacote (Sigma-Aldrich) to prevent flies from walking on the ceiling, and kept under a fume hood to dry for at least 10 min.

The floor of the chamber was tiled with 16 microphones (Knowles NR-23158) that were embedded into a custom-made PCB board (design modified from[27]). The microphones were covered with a thin, white paper for the flies to walk on and to record sound and vibration. Microphone signals were amplified using a custom-build amplifier[50] and digitized using a data acquisition card (National Instruments PCIe-6343) at a sampling rate of 10 kHz.

Fly behavior was recorded from above using a USB camera (FLIR flea3 FL3-U3-13Y3M-C, 100 frames per second (fps), 912 × 920 pixels), equipped with a 35 mm f1.4 objective (Thorlabs MVL35M1). The chamber was illuminated with weak blue light (470 nm) and white room light. For optogenetic experiments, the room light was turned off to reduce interference between illumination and activation wavelengths. A 500 nm shortpass filter (Edmund Optics, 500 nm 50 mm diameter, OD 4.0 Shortpass Filter) filtered out green (525 nm) and red (625 nm) wavelengths used for optogenetics.

To match the males' abdominal quivering with the vibration pulses recorded on the microphones, we recorded videos with higher spatial (1200 × 1200 pixel frames covering a chamber with diameter 11 mm) and temporal (150 fps) resolution. The chamber was centered on one of the 16 recording microphones and illuminated with white LEDs.

Synchronized recordings of audio, video, and delivery of optogenetic stimuli were controlled using custom software (https://janclemenslab.org/etho).

As a control, we also measured the substrate deflections induced by vibrations using a PSV-400 laser Doppler vibrometer (Polytec GmbH) in the same chamber and paper substrate used above. The laser

**Table 1 | Fly lines used**

| Figures | Name | Genotype | Reference | Provided by |
|---|---|---|---|---|
| 1 - 3, 5, 6, S1 - S4, S7 | wild-type | *Drosophila melanogaster* NM91 | 27 | Peter Andolfatto |
| S1D | wild-type | *Drosophila melanogaster* OregonR | | |
| 3, S4 | vGlut | VGlut1[OK371]-GAL4/+; UAS-GtACR1.d.EYFP(attP2)/+ | 54 | vGlut by Martin Göpfert |
| 3, S4 | DNp28 | 20xUAS-IVS-CsChrimson.mVenus(attP40)/R11H10-p65.AD(attP40); | 55,56 | DNp28 (SS01587) by Gwyneth Card |
| | | VT033947-GAL4.DBD(attP2)/+ | 99 | CsChrimson by André Fiala |
| 4, 5, S5, S7 | pC2l | UAS(FRT.STOP)CsChrimson.mVenus(attp14)/+; | 24 | Vivek Jayaraman |
| | | GMR42B01-Gal4(attP2)/8xLexAop2-FLP(attp2),dsx-LexA | | |
| 4, 5, S5, S7 | P1a | 20xUAS-IVS-CsChrimson.mVenus(attP40)/R15A01-p65.AD(attP40); | 64 | David Anderson |
| | | R71G01-GAL4.DBD(attP2)/+ | | |
| 4 | pMP2 | 20xUAS-IVS-CsChrimson.mVenus(attP40)/VT026873-p65.AD(attP40); | 47 | Joshua Lillvis |
| | | VT028160-GAL4.DBD(attP2)/+ | | |
| 4 | pIP10 | UAS(FRT.STOP)CsChrimson.mVenus(attp14)/+; VT40556-GAL4/fru-FLP | 23 | |

beam was directed through the transparent lid perpendicular to the paper surface at a distance of 1-4 mm near a stationary male courting a female (Fig. S1). Data obtained with the laser vibrometer were high-pass filtered (Butterworth, 60 Hz) before analysis.

### Behavioral assays
For all experiments, 3 to 7-day-old naive males and virgin females were used. Flies were introduced gently into the chamber using an aspirator. All recordings were performed during the flies' morning activity peak and started within 120 min of the incubator lights switching on. Recordings of video and audio were performed for 30 min in the regular chamber, for 10 min in the smaller chamber, and for 2 min during laser vibrometry. In experiments using males with amputated wings (Fig. S1G–H), flies were cold-anesthetized and both wings were cut using fine scissors at least 18 h before the experiment.

To induce sexual satiation (Fig. 5H–N), males were transferred individually into food-containing vials with 10–15 virgin females and allowed to freely interact and copulate for 4–6 h. The control males came from groups of 10–15 males with the same genotype (pC2l-CsChrimson or P1a-CsChrimson). After the pre-exposure period, all flies were quickly anesthetized on ice to separate one male from the group, who was gently transferred into an empty vial to recover for 15 min. Then he was gently introduced into the behavioral chamber, and the optogenetic activation experiment was started.

### Optogenetics
Flies were kept for at least 3 days prior to the experiment on food containing retinal (1 ml all-trans retinal (Sigma-Aldrich) solution (100 mM in 95% ethanol) per 100 ml food). To prevent the degradation of the retinal and continuous neural activation, the vials were wrapped in aluminum foil. Control flies were either parental controls (Figs. 3, 4) or had the same genotype as experimental flies and were kept on regular food without additional retinal. Note that regular food contains trace amounts of retinal, and drivers with strong expression can therefore produce effects even in the non-retinal controls.

For neural inactivation, we used the GtACR1 channel[54,98], which was excited using a green LED (625 nm). For inactivation of vGlut (Fig. 3A–B) we used an LED intensity of 14 mW/cm². This intensity was chosen in pilot experiments to induce reliable female stationarity with minimizing off-target effects from the light stimulus in male courters. Experiments consisted of 40 trials of optogenetic stimulation. Each trial started with 5 s stimulation (green LED on), followed a pause of 25 s. For neural activation, we used the CsChrimson channel[99], which was activated using a red LED (625 nm). For activation of DNp28 (Fig. 3C–D) we used an LED intensity of 89 mW/cm². Each experiment consisted of 30 trials of optogenetic stimulation. Each experimental trial started with 5 s stimulation followed by a pause of of 25 s. For activation of pIP10 and pMP2 (Fig. 4B) we used LED intensities of 1, 2, 3, 5 and 14 mW/cm² presented in order. Each experiment consisted of 15 trials. Each trial started with 5 s of optogenetic stimulation followed by pause of 115 s. For pC2l and P1a activation (Figs. 4–5) we used LED intensities 14, 27, 83, and 209 (P1a only) mW/cm². In subsequent experiments for which only a single intensity was used, we selected the lowest intensity that evoked reliable vibration (27 mW/cm² for pC2l and P1a). Each experiment consisted of 7 trials of optogenetic stimulation, and each trial started with 5 s of optogenetic stimulation followed by a pause of 120 s.

### Analysis of microphone signals
Multimodal courtship signals (pulse, sine, vibration) were manually annotated using the graphical user interface of DAS[100]. For optogenetic manipulation of female walking (Fig. 3) and the satiation assay (Fig. 5H–N), the annotators were blind to the experimental condition.

*Pulse and vibration trains* were defined as groups of pulses with an interval less than 2–2.5 times the modal interval (80 ms for pulse song, 400 ms for vibration). The *signal fraction* is the fraction of all courtship frames in which a specific signal–pulse, sine, or vibration–was produced.

*Transition probabilities* between signals correspond to the fraction of signals of a given type that were followed by a given other signal (i.e., fraction of pulse trains followed by sine song, or pulse song, or vibrations), regardless of the duration of the silent pause between trains. We then averaged the transition probabilities over all 14 pairs of NM91 wild-type flies.

*Signal probabilities* for experiments with optogenetic neural activation or inactivation are given as the fraction of trials during which sine song or pulse and vibration trains were produced. We then computed the mean across trials pooled across all males. For experiments with speed-controlled females (Fig. 3) and with optogenetic activation of P1a and pC2l in males paired with a female (Fig. 5C–F), we only considered time points during which the male courted the female.

*Signal sequences* were analyzed in optogenetically manipulated solitary males by pooling data across all activation intensities. Starting from stimulus onset or stimulus offset, we determined which signal the males started with and which signal types followed. Pauses between signal trains lasting longer than 0.5 s were considered "no signal".

### Behavioral data analysis
Flies were tracked using standard procedures (estimation of background as the median frame, subtraction of background from each frame, thresholding, and localization of flies using a Gaussian mixture model). The location of individual body parts (head, thorax, abdomen,

**Table 2 | Open source software used**

| Resource | Link (citation) |
|---|---|
| DeepPoseKit | https://github.com/jgraving/DeepPoseKit[101] |
| DeepAudioSegmenter | https://github.com/janclemenslab/das[100] |
| GLM utilities | https://github.com/janclemenslab/glm_utils |
| Inkscape 0.92 | https://inkscape.org |
| Python 3.7–3.12 | https://python.org |
| scikit learn | https://scikit-learn.org[103] |
| seaborn | https://seaborn.pydata.org[113] |
| SLEAP | https://sleap.ai[51] |
| xarray-behave | https://github.com/janclemenslab/xarray-behave |
| etho | https://github.com/janclemenslab/etho |
| pandas | https://pandas.pydata.org[114] |
| numba | https://github.com/numba/numba[115] |
| networkx | https://networkx.org[109] |
| navis | https://navis-org.github.io/navis[110] |
| natverse flybrains | https://natverse.org[116] |
| flywire codex | https://codex.flywire.ai[105] |

left and right wing) were then tracked using DeepPoseKit[101]. For most analyses, the tracking data was downsampled from the original frame rate of 100 Hz (fps) to 50 Hz. All time points after the beginning of copulation were excluded from analysis.

To show traces of signal probabilities or velocities for optogenetic experiments or onset/offset analysis (Figs. 3–5, S4–5), we pooled data across flies and computed the mean (for signal probabilities) or median (for velocities) across stimulation trials or onsets and offsets. To eliminate tracking errors from velocity or wing angle data, we excluded data points where the distance between male and female thoraces dropped below 1 mm and were the tracking confidence for the head or thorax was less than 50%. All traces shown for optogenetic experiments (Figs. 3–5) are smoothed with a Gaussian window with a standard deviation of 0.1 s, velocity traces were smoothed with a Gaussian window with a standard deviation of 1 s.

*Courtship* was defined as time points during which the male was within 8 mm (6 mm for GLM analysis) of the female and ± 60° behind her. The *courtship index* is the fraction of time points that are courtship from the beginning of the recording until copulation started or the recording ended.

### Correlating abdominal quivering and vibration pulses

Flies positions and body parts in the high-resolution videos were tracked using SLEAP[51]. We then independently annotated abdominal quivering in the video, visible in the top-down view as a brief shortening of the abdomen, and vibration pulses in the audio.

### Behavioral modeling

Multinomial Generalized Linear Models (GLMs) were used to identify the behavioral cues and contexts that drive the choice between song (pulse, sine) and vibration. Models were fitted to predict whether the male produced a song, vibration or no signal at any moment in time.

As behavioral cues, we extracted 19 metrics from the fly tracks of 14 male-female pairs of NM91 using xarray-behave (Table 2): male or female rotational speed, rotational acceleration, velocity and its forward and lateral components, acceleration and its forward and lateral components, male-female distance, as well as the male's relative angle (male position relative to female body axis) and relative orientation (males heading relative to female center). We only considered courtship frames and frames before copulation.

The cues for each pair were z-scored and then pooled across pairs. That way, each GLM was fitted to the data from multiple pairs. Since we

were interested in identifying the time course of each cue that best predicted signaling, we delay-embedded the cues. That is, the signals in each time point were predicted using the time course of each cue in the 1 s preceding that time point. To reduce dimensionality, we projected each 1 s onto a basis of four raised cosines covering the 1 s time window with logarithmic spacing[102]. Thereby, the cues' time course in the 1 s preceding each time point was predicted by 4 values. The temporal filters (Fig. 2H) were recovered from the 4 weights learned by the GLM by back-projecting the raised cosine basis to time. The filter sum (Fig. 2G, S3B) was given by the sum of all filter values in the time domain.

Since the fraction of song, vibration and no signal in the data was skewed towards no signals, we balanced the data prior to fitting by randomly sub-sampling an equal number from each prediction target (song, vibration, no signal). This yielded 73,562 time points per signal type as inputs to the model fitting.

**GLM fitting and evaluation.** Data points of behavioral cues were split into 90% training data and 10% test data. Each model was fitted 10 times, each time with random train-test splits and balancing. Models were fitted using `LogisticRegressionCV` from scikit-learn[103], with L2 regularization, ten-fold cross-validation and a maximum of 500 iterations.

The performance of each fitted model was quantified by comparing model predictions on the test set to behavioral groundtruth data. Predicted and true signals were tabulated in a confusion matrix, normalized by the true signals (Fig. 2C, E). Diagonal matrix elements correspond to correct predictions (plotted in Fig. 2F) and off-diagonal elements correspond to prediction errors. To obtain a single score of the performance, we computed the accuracy as the average over the diagonal values. We fitted two types of models: To assess the general ability of the cues to predict the males' signal choice, we fitted a model that used all 19 cues (Fig. 2C). As a second step, to assess to information contributed by each individual cue, while all other cues were shuffled, we fitted separate models for each cue and assessed their performance (Fig. 2D).

### Connectome analyses

Connectome analyses in Fig. S9 were based on the female whole brain connectome, flywire[74,75,104], since no male brain connectome data is currently available. The data was downloaded from flywire codex (https://codex.flywire.ai/api/download, v783)[105] and further processed using open source packages (see Table 2). The pC1 and pC2 neurons were identified based on existing cell-type annotations in flywire[104] and connections[106–108] were identified using the `all_simple_paths` function of the networkx package[109]. The outline of the brain and the neuronal skeletons were plotted using navis[110] and natverse's flybrains package[111].

### Circuit model

**Model structure and working principle.** The primary goal of the model is to synthesize the experimental results and show that our current model of the circuit is sufficient to explain the behavioral data. The model is well supported by existing and our own data, and consists of four main components:

1. The social cue integrating neuron groups P1a and pC2l mediates acute effects of activation via connections to descending command-like neurons.
2. A recurrent neural network (RNN) downstream of P1a mediates the long-term effects of circuit activation.
3. Two descending command-like neurons, pIP10 and DNvib, drive song and vibration in the ventral nerve chord.
4. Mutual inhibition between or downstream of pIP10 and DNvib reduces the overlap between song and vibration.

P1a and pC2l have been shown to be activated by social cues in numerous studies. The pC2l neurons are activated by male pulse

song[24] and likely also visual[61] and other cues. The P1a neurons receive inputs from volatile and contact chemical cues[45,62,63]. Our behavioral results leave open the possibility that additional, still unidentified cues activate P1a.

In our experiments, activation of P1a and pC2l drove vibration and song, respectively, with short latency (Fig. 4). This suggests that they have short connections spanning only one or a few synapses to command-like descending neurons. Direct connectivity between pC2l and the song DN pIP10 has been established anatomically and functionally[60]. Short connections between P1a and descending command neurons are not known but are likely, given the behavioral data. This connection can be tested directly once DNvib has been identified.

Vibrations were also driven at the offset of pC2l. In the model, this is mediated via a pC2l to P1a connection (Fig. S8B, E). pC2l activity would induce relatively weak and slowly decaying activity in P1a. A pC2l to P1a connection has been hypothesized in a recent paper on song patterning[33] and was required to explain the production of complex song upon pC2l activation. Our data provides independent support for such a connection. The activity of P1a has been shown to decay slowly with a time constant of 5–10 s[63], which matches the time constant of the offset vibrations after pC2l activation (Fig. 4). This supports the idea of offset vibrations after pC2l activation being driven by this slowly decaying P1a activity.

An RNN downstream of P1a maintains vibration activity for tens of seconds. Elements of the RNN have been characterized previously using behavioral and imaging experiments, and the pCd neurons are members of this network[31]. Connectivity downstream of the RNN is unknown. For simplicity, we assume that the RNN drives both song and vibration DNs. However, alternative implementations are possible. Signaling after P1a activation in solitary males is strongly biased towards vibrations, and this is reflected in stronger relative connectivity from the RNN to the DNvib versus pIP10 in our model.

Lastly, mutual inhibition downstream of P1a and pC2l reduces the overlap between song and vibration, and induces switching between song and vibration during the persistent phase driven by adaptation and noise. This component of the model is derived from models of bistable phenomena[69]. Mutual inhibition could be implemented at different stages downstream of P1a and pC2l: Upstream of pIP10 and DNvib, between pIP10 and DNvib, or downstream of the DNs in the VNC. For simplicity, we model mutual inhibition as happening between pIP10 and DNvib. pIP10 receives input from pC2l and the RNN, and DNvib receives input from P1a and the RNN. Both DNs adapt, which is supported by the observation of spike-frequency adaptation in patch clamp recordings of pIP10[33]. pIP10 activity drives song in the VNC and an interneuron that inhibits DNvib. DNvib activity drives vibrations in the VNC and an interneuron that inhibits pIP10. The latter interneuron adapts, which acts as a high-pass filter that speeds up the inputs from P1a-DNvib to account for the short latency of inhibition of song upon P1a activation (Fig. 5). Gaussian noise is added to the output of pIP10 and DNvib to enable stochastic switching between song and vibration in the persistent phase.

Since we were interested in circuit dynamics on a timescale of seconds, we implemented a rate-based model, in which the activity of individual neurons is represented by continuous variables that are considered to be proportional to the firing rate of the cell (individual cells, e.g., for pIP10, or cell clusters, e.g., P1a or pC2l). To translate the activity of pIP10 and DNvib to behavior, we consider their activity to be proportional to the probability of observing song and vibration, respectively. Trial averaged plots show the average probability over 100 model simulations with different noise patterns. Mathematical details of the circuit model are specified in the Supplementary information. Code for running the model can be found at https://github.com/janclemenslab/vibmodel[112].

## Statistical analyses

All tests were Wilcoxon (for paired data) or Mann-Whitney-U tests (for unpaired data). The significance levels for multiple comparisons were adjusted from 0.05 using the Bonferroni method. For assessing the effect of optogenetic activation in courting males, statistics only include males that intensely courted the female 10 s before and during optogenetic activation. Intense courtship was defined as a courtship index of 0.9 (see above).

## Reporting summary

Further information on research design is available in the Nature Portfolio Reporting Summary linked to this article.

## Data availability

All data supporting the findings of this study are available within the paper, its Supplementary Information or a public repository. Source data are provided with this paper. Raw experimental data generated in this study have been deposited in the Göttingen Research Online database (https://doi.org/10.25625/4R4MZU). Source data are provided with this paper.

## Code availability

All data analyses were performed using the software listed in Table 2. Code for running the computational model is deposited at https://github.com/janclemenslab/vibmodel[112].

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

## Acknowledgments

We thank Tina Zahrie, Jannis Hainke, Maximilian Ferle, Karla Rivera, Alina Seidel for help with annotation and data acquisition, Frank Kötting, Stephan Löwe from the ENI workshop for help with designing behavioral chambers, Gesa Hoffmann, Jan Schöning, Christine Gündner, Christiane Becker for technical and adminstrative assistance. Martin Göpfert and Philip Hehlert provided access to a laser vibrometry setup. Gwyneth Card, David Anderson, Vivek Jayaraman, André Fiala, Peter Andolfatto, Joshua Lillvis, Martin Göpfert, Janelia flylight, Bloomington stock center for gifts of flies. We thank all members of the Clemens lab as well as Frederic Roemschied, Daniela Vallentin, Mala Murthy, and Xinping Li for feedback on the manuscript. We thank the de Bivort Lab for making their fly clip art publicly available. This work was funded via an Emmy Noether Grant (Project number 329518246) and an ERC Starting Grant (Grant agreement No. 851210) to JC.

## Author contributions

Conceptualization - E.S., A.K and J.C. Animals and behavioral experiments - E.S., A.K., M.S., B.S., S.R. and K.A. Modeling and analysis - E.S. and J.C. First draft - E.S. and J.C. Feedback on draft - A.K., M.S., B.S., S.R. and K.A.

## Funding

## Competing interests

The authors declare to have no competing interests as defined by Nature Portfolio or other interests that might be perceived to influence the interpretation of the article.
