## [Transparent Peer Review file · Nature Communications]

A neural circuit for context-dependent multimodal signaling in *Drosophila*

Corresponding Author: Professor Jan Clemens

Version 0:

Reviewer comments:

Reviewer #1

(Remarks to the Author)

This paper by Steinfath and colleagues presents a compelling investigation into the multimodal signaling strategies employed by *Drosophila* during courtship. The authors have developed a novel and valuable experimental setup that allows for the isolation and analysis of distinct courtship signals, specifically song (sine and pulse) and vibrations. This tool alone represents a significant contribution to the field.

The study provides strong evidence that vibrations play a more prominent role in courtship than previously thought, occurring twice as often as song and rarely overlapping with it. The authors meticulously demonstrate that transitions between song and vibration are dependent on female locomotion and distance, challenging the view of males as passive during courtship and instead highlighting the importance of the vibration stage. The use of optogenetic manipulation of females to induce only stopping behavior, reliably triggering vibrations, is a particularly elegant approach that strengthens their conclusions by effectively ruling out other potential influencing factors. The investigation into the neural circuitry underlying these behaviors is equally impressive. Optogenetic manipulation of P1a and PC2l neurons within the song circuit revealed distinct roles for these neurons in eliciting song and vibration, with P1a driving long-lasting vibrations and PC2l primarily driving song followed by vibrations after stimulation offset. The authors' observation that P1a activation leads to both male stopping and probabilistic triggering of vibrations, while described sequentially, could benefit from a bit more discussion regarding the potential mechanistic link between these two outputs.

The study's strength lies in its series of well-designed experiments that progressively address key questions. The first "smart move" – examining the interplay of P1A and PC2l in a social context with a female – revealed a mutual inhibition mechanism that prevents signal overlap. The second "smart move" – repeating the experiment during courtship interaction – showed a bias towards vibration upon neuron activation, but also demonstrated the influence of external cues in overriding isolated circuit dynamics, naturally leading to the question of what these cues are. The third "smart move" – investigating the role of motivational state by using sexually satiated males – demonstrated a strong decrease in both vibration and song after neuron activation, further enriching the model.

Finally, the authors present a computational network model that effectively replicates their experimental findings. This model provides a valuable foundation for future studies exploring the descending pathways and testing further predictions.

The data is clearly presented in well-conceptualized figures with effective color schemes. The text is generally well-written, although some minor repetition in wording ('understood', 'unknown' reoccur frequently) could be addressed. The methods and mathematical model are thoroughly described. Overall, this is a very strong paper that makes a significant contribution to our understanding of *Drosophila* courtship behavior. Therefore, I recommend it to be published in Nature Communications.

Minor Comments:

- The abstract could be improved by focusing on a more concise narrative. Specifically, I suggest removing the third sentence as it currently stands, as it mentions one hypothesis without its alternative, making it slightly confusing.
- The order of presentation of "no signal," "vibration," and "song" in the subpanels and figure legends of Figure 2 (B, C, and E) should be made consistent for clarity.
- Regarding the behavior of the GtACR1 expressing females used in the experiments: Do they exhibit any abnormal behaviors (related to locomotion or other courtship cues) that might influence the male's behavior?

- Figure 4: pMP2 is mentioned in the legend but not shown in the figure. This discrepancy needs to be addressed. Also, the units used (males, fly or absent) should be unified throughout the figure and its legend.
- Line 340, 'its transmission'

Reviewer #2

(Remarks to the Author)

Steinfath and Khalili et al., in this manuscript explore the neural circuit mechanisms underlying multimodal actions (songs vs. vibrations) in *Drosophila*. This study characterizes the dynamics of these two distinct, non-overlapping, behavioral modes during natural courtship. Subsequently, using a combination of cell-type specific perturbations and computational modeling, the authors identify the core elements of a shared multimodal circuit for songs and vibrations.

Overall, the study is timely, interesting, and well-executed. The experimental logic is sound and the manuscript is well written. This article sets the foundation for detailed investigation into circuit mechanisms of behavioral mode switching in *Drosophila*.

Here are some suggestions, in no particular order, to improve the manuscript –

1. Throughout the manuscript, the two behavioral modes (songs and vibrations) are referred to as "signals". To me, the framing of these two modes as "signals" assumes that the receiver --- a female in this case --- is constantly evaluating and acting upon the signal. While this may be the case, it is not guaranteed to be so. For example, wingless males without any songs still mate with equal success. Therefore, it may be more parsimonious to be agnostic about the intended consequences of these modes, and without any loss of generality, simply refer to them as "actions" --- which is what they are.
2. In Fig 1B-E, the authors identify three behavioral states: sine, pulse, and vibrations. In F and G, however, the authors also plot the probability of songs (0.14) independently and the total probability adds up to 1.14. This is technically wrong and potentially confusing. Since sine + pulse probabilities also add up to be 0.14, I believe the authors simply combined the pulse and sine modes to define the song. The authors might want to clarify this issue and either plot sine vs. pulse vs. vibration or song vs. vibration.
3. In Fig. 2C, I am surprised and a bit worried that the true "no signal" category is incorrectly predicted to be a signal (song or vibration) majority of times (54%). This seems to be even higher for all three panels of Fig. 1E as well. Shouldn't "no signal" be the easiest state to classify? Does this not raise concerns about whether we can trust the GLM output for songs and vibrations?
4. In line 105, the authors write that "With only rare confusions between song and vibration we were able to determine that feedback 106 cues determine the choice between song and vibration (Fig. 2C)." How is Fig. 2C showing that feedback cues are important?
5. Since Fig. 2E plots accuracy of predictions, calling it a confusion matrix in the legend is a bit confusing.
6. In Line 111, the authors write "These findings indicate that male and female movement patterns are the strongest predictors of vibration production during courtship." What could have been a viable alternative hypothesis that is ruled out here?
7. In Line 133, the authors write "Having access to vibrations during courtship, we found that part of this 'passive' 133 state is not idle, but that the male actively signals to the female". Similar to point 1 above, how do we know whether it is an advertisement signal.
8. Since GLMs can only identify correlations between behavioral variables, I applaud the authors for manipulating female locomotion state to infer its causal effect on male's behavior. In line 158, the authors write "It is therefore likely that the male's locomotor state controls the choice between song and vibration, and is not influenced by the female movement." I agree with the authors but isn't this contradictory to the main message of Fig 3, whose caption says "Female immobility is a necessary and sufficient trigger for male vibrations." The authors should clarify their conclusion for this section.
9. In Figure 4A, it will be helpful to include the pMP2 neuron as well. In line 199: the authors write "Interestingly, at the offset of strong activation, we observed vibrations lasting 5–10 s 200 (Fig. 4F). pC2I neurons are known to produce sine song at activation offset [7, 10, 30] but this sine song is much shorter." So, is it pulse -> sine -> vibration? What fraction of times does the pulse state transition to sine en route to vibrations?
10. In line 220, the authors write that "60% of the stationary males vibrated independent of activation strength (Fig. 4J), suggesting that P1a neurons do not induce a drive to vibrate which in turn stops males." This is clever and this kind of probabilistic transition probabilities will be helpful for the sine, pulse, and vibrational modes as well. The authors should consider showing the transition probabilities between all of these states for the optogenetic manipulation experiments.

11. Given that there is known mutual inhibition between the sine and the pulse modes within the "song" state, situating the circuit for mutual inhibition between songs and vibrations within that context will be very helpful.

12. In Fig. 4E, about half of the magenta (song) trials are followed by green (vibration). This is close to 50% transition probability. In natural behavior (Fig 1), this transition is very rare. The authors discuss the potential circuit state differences between natural behavior vs. artificial activation with optogenetics. One additional confound is the potential off-target effects of acute perturbations (Otchy et al, 2015). Driving neural activity beyond the natural regime in recurrent circuits might overestimate causal effects. What do the authors think of this possibility? What is known about the neural activity of P1a neurons during vibrations?

13. In Fig 5A-B, the blue inhibitory connection is hypothesized. Please highlight this clearly in the figure panel to avoid confusion.

14. In Fig 5K-N, consider staggering the "control" vs. "satiated" data points. Currently, it is difficult to discern.

15. In line 279, the authors claim "we now show that pC2I neurons were also subject to motivational control implying a global effect of motivation on the courtship circuit." I am not convinced of this claim. The difference between the Song probabilities in satiation vs. control (purple lines vs. purple shaded area) upon pC2I during the actual stimulation epoch is quite small. The major difference between the two contexts is observed during stimulation offset. But couldn't offset responses be indirectly due to the context-dependence of P1a neuron or the RNN or the walking drive circuit?

16. In line 301, authors write that "Activation of the RNN requires strong and direct activation of P1a neurons and mediates the long-term persistence of multimodal signals via connections to the descending neurons for song and vibration." Is this known from experiments or is this an assumption of the model? In general, this paragraph may benefit from clarifying which of the model characteristics are derived from previous experimental facts vs. hypothesized to be the case.

Reviewer #3

(Remarks to the Author)

This is an exciting and important work which expands our understanding of *Drosophila* courtship behavior, and identifies neural circuit motifs for the integration of multimodal behaviors. The authors develop a novel behavioral assay to simultaneously quantify vibrational and acoustic (song) communication occurring during courtship. Precise behavioral analysis reveals that the absence of locomotion, stationarity, is highly coordinated with vibration while movement is highly correlated with song. Behavioral overlap between vibrational and acoustic cues is rarely observed which suggested these behaviors could be coordinated by a single circuit.

Through a series of optogenetic experiments the authors identify separate premotor pathways responsible for both song (mediated through PC2I) and vibration (mediated through P1a) and evidence for mutual inhibition between the two pathways downstream of P1a and PC2I. In addition, they provide evidence descending pathways from P1a can also coordinate vibration and location. Based on these results the authors propose several models which they then elegantly test with additional behavioral and optogenetic manipulations. Ultimately, this work provides a compelling circuit dissection of multimodal elements of courtship behavior with elements downstream of P1a driving vibration and elements downstream of PC2I, driving song. The authors further demonstrate that coordination between the two behaviors can be explained by mutual inhibition. These results are supported by natural courtship behaviors, optogenetic manipulations and modeling.

The question is of broad interest, the methods are innovative and the conclusions are well supported by the data, making the work well suited to publication in *Nature Neuroscience*.

Major comments:

- Did the authors test if any features of the vibration behaviors, e.g. timing, duration, onset are influenced by the substrate? As this is an artificial setting, I would be curious to see how vibrations may be altered if the substrate was more similar to conditions of natural courtship (e.g. fermenting organic matter or fruit). While this is not necessary to reproduce here, it would be interesting to see how stereotyped the behavior is if the transmission of the vibration is changed by the substrate.

Minor comments:

- Methods: Could the authors please comment on how the LED intensities for optogenetic activation were chosen when only a single intensity is used.
- Figure 5C,D: The schematic of the courting flies seems a bit misaligned and distracts from the figure. Please reposition the schematic or consider removing it.
- Figure 6A: PC2I is missing the l.
- Figure 6 legend: In the first sentence a space is missing between "excitatory and inhibitory"
- Figure S9 is embedded in the middle of the references.
-

Version 1:

Reviewer comments:

Reviewer #1

(Remarks to the Author)

The authors have done an excellent job of rewriting the abstract and addressing all of my minor concerns. I do not have any further comments and strongly support the acceptance of this manuscript for publication in Nature Communications. This is an excellent piece of work.

(Remarks on code availability)

Reviewer #2

(Remarks to the Author)

The authors have done a wonderful job in addressing my concerns. I support the publication of the manuscript in Nature Communication.

(Remarks on code availability)

Reviewer #3

(Remarks to the Author)

My comments have been addressed and I thank the authors for the detailed responses. I find the manuscript much improved by the author's response to all Reviewers and recommend the manuscript for publication.

(Remarks on code availability)

We thank the Reviewers for their positive and constructive feedback. We were gratified that all Reviewers found the work to be of broad interest and our methods and data sets to be both thorough and impressive. Here we provide point by point responses to the Reviewer comments. We have made significant changes to the manuscript in response to the Reviewers' comments. These include:

- 1) A more nuanced assessment of whether song and vibration can be considered signals.
- 2) Modified the text and several figure panels (Figs. 1F, 4A, 5A-B and 5K-N, S5G-J, S6) to improve clarity.
- 3) New analyses regarding transitions between multimodal signals (new Figs S5G-J).
- 4) New analyses strengthening our argument regarding the coordination of locomotion and multimodal signaling (new Fig. S6).

REVIEWER COMMENTS

Reviewer #1 (Remarks to the Author):

This paper by Steinfath and colleagues presents a compelling investigation into the multimodal signaling strategies employed by Drosophila during courtship. The authors have developed a novel and valuable experimental setup that allows for the isolation and analysis of distinct courtship signals, specifically song (sine and pulse) and vibrations. This tool alone represents a significant contribution to the field.

The study provides strong evidence that vibrations play a more prominent role in courtship than previously thought, occurring twice as often as song and rarely overlapping with it. The authors meticulously demonstrate that transitions between song and vibration are dependent on female locomotion and distance, challenging the view of males as passive during courtship and instead highlighting the importance of the vibration stage. The use of optogenetic manipulation of females to induce only stopping behavior, reliably triggering vibrations, is a particularly elegant approach that strengthens their conclusions by effectively ruling out other potential influencing factors. The investigation into the neural circuitry underlying these behaviors is equally impressive. Optogenetic manipulation of P1a and PC2l neurons within the song circuit revealed distinct roles for these neurons in eliciting song and vibration, with P1a driving long-lasting vibrations and PC2l primarily driving song followed by vibrations after stimulation offset. The authors' observation that P1a activation leads to both male stopping and probabilistic triggering of vibrations, while described sequentially, could benefit from a bit more discussion regarding the potential mechanistic link between these two outputs.

The study's strength lies in its series of well-designed experiments that progressively address key questions. The first "smart move" – examining the interplay of P1A and PC2l in a social context with a female – revealed a mutual inhibition mechanism that prevents signal overlap. The second "smart move" – repeating the experiment during courtship interaction – showed a bias towards vibration upon neuron activation, but also demonstrated the influence of external cues in overriding isolated circuit dynamics, naturally leading to the question of what

these cues are. The third "smart move" – investigating the role of motivational state by using sexually satiated males – demonstrated a strong decrease in both vibration and song after neuron activation, further enriching the model.

Finally, the authors present a computational network model that effectively replicates their experimental findings. This model provides a valuable foundation for future studies exploring the descending pathways and testing further predictions.

*The data is clearly presented in well-conceptualized figures with effective color schemes. The text is generally well-written, although some minor repetition in wording ('understood', 'unknown' reoccur frequently) could be addressed. The methods and mathematical model are thoroughly described. Overall, this is a very strong paper that makes a significant contribution to our understanding of *Drosophila* courtship behavior. Therefore, I recommend it to be published in *Nature Communications*.*

Minor Comments:

- The abstract could be improved by focusing on a more concise narrative. Specifically, I suggest removing the third sentence as it currently stands, as it mentions one hypothesis without its alternative, making it slightly confusing.*

We thank the reviewer for this suggestion. We have re-written the abstract:

*Many animals produce multimodal displays that combine acoustic, visual, or vibratory signals, yet the neural mechanisms coordinating these behaviors remain unclear. Using *Drosophila* courtship as a model, we reveal how a single neural circuit integrates sensory cues and motivational state to orchestrate multimodal signaling. Male flies produce both airborne song and substrate-borne vibrations during courtship, but in distinct, largely non-overlapping contexts. We demonstrate that the same brain neurons that drive song also control vibrations through separate pre-motor pathways, with cell-type specific dynamics. This shared circuit coordinates multimodal displays with locomotion, ensuring vibrations are produced only when they can effectively reach the female. The circuit employs shared motifs - recurrence and mutual inhibition - that enable dynamic control of multimodal signals by external cues and internal state. A computational model confirms that these motifs are sufficient to explain the observed behavioral dynamics. Our findings illustrate how simple neural circuit elements can be combined to select and coordinate complex multimodal behaviors.*

- The order of presentation of "no signal," "vibration," and "song" in the subpanels and figure legends of Figure 2 (B, C, and E) should be made consistent for clarity.*

We thank the reviewer for pointing out this inconsistency. The order of presentation is now consistent across all figure panels and in the caption.

- Regarding the behavior of the *GtACR1* expressing females used in the experiments: Do they exhibit any abnormal behaviors (related to locomotion or other courtship cues) that might influence the male's behavior?*

In these experiments, we focused on comparing male vibration behavior outside and during activation (DNp28-Chrimson) and inactivation (vGlut-GtACR1) of neurons that control female locomotion and our experiments show clear effects of optogenetic manipulation on male vibration behaviors. Outside of optogenetic stimulation, females behaved normally apart from

a weak reduction in walking speed for GtACR1 females. Males courted and sang to females of all genotypes vigorously, suggesting that any small behavioral differences between females did not affect the vibration behavior we thought to manipulate.

- *Figure 4: pMP2 is mentioned in the legend but not shown in the figure. This discrepancy needs to be addressed. Also, the units used (males, fly or absent) should be unified throughout the figure and its legend.*

We have added pMP2 to the schematic in Fig. 4A and unified all units to “flies” and clarified the N for all panels.

- *Line 340, ‘its transmission’*

Fixed.

Reviewer #2 (Remarks to the Author):

Steinfath and Khalili et al., in this manuscript explore the neural circuit mechanisms underlying multimodal actions (songs vs. vibrations) in Drosophila. This study characterizes the dynamics of these two distinct, non-overlapping, behavioral modes during natural courtship. Subsequently, using a combination of cell-type specific perturbations and computational modeling, the authors identify the core elements of a shared multimodal circuit for songs and vibrations.

Overall, the study is timely, interesting, and well-executed. The experimental logic is sound and the manuscript is well written. This article sets the foundation for detailed investigation into circuit mechanisms of behavioral mode switching in Drosophila.

Here are some suggestions, in no particular order, to improve the manuscript –

- 1. Throughout the manuscript, the two behavioral modes (songs and vibrations) are referred to as "signals". To me, the framing of these two modes as "signals" assumes that the receiver --- a female in this case --- is constantly evaluating and acting upon the signal. While this may be the case, it is not guaranteed to be so. For example, wingless males without any songs still mate with equal success. Therefore, it may be more parsimonious to be agnostic about the intended consequences of these modes, and without any loss of generality, simply refer to them as "actions" --- which is what they are.*

We thank the reviewer for raising this important point. When writing the manuscript, we also discussed whether we have enough evidence to call both song and vibration signals. According to the common definition of "signal" in the animal communication field: *A signal is something "done" by an animal (the sender) that influences another animal's (the receiver's) behavior.* We have sufficient evidence that both song and vibration influence the behavior of the female receiver, and therefore we decided to call both song and vibration "signals."

Song has been shown to be crucial for female mating decisions and affects female behavior in at least three ways:

1. Song reduces female walking speed (van Schilcher 1976, Bussell 2014, Deutsch 2019).
2. Song promotes receptivity in virgin females, leading to vaginal plate opening that triggers male copulation attempts (Wang 2021). Removing song by deafening females or muting males reduces copulation rates in many assays (Coen 2014, McKelvey 2021, Wang 2021).
3. Song triggers rejection behaviors, like ovipositor extrusion in recently mated, and therefore non-receptive, females (Mezzera 2020).
4. Song also affects male behaviors, triggering courtship (male chaining - Eberl 2000) or aggression (Sten 2025).

These clear and reproducible effects of song on female and male behaviors demonstrate that song is indeed a signal.

Vibrations have received much less attention in the field, and the evidence of vibrations affecting female behaviors is less clear than for song. However, several studies report behavioral effects of vibrations in females: First, like song, vibrations slow or stop females (Fabre 2012). While our results suggest that female slowing drives vibrations, this is not contradictory, since vibrations might prolong phases of female stationarity. Second, vibrations affect female mating decisions. Females detect vibrations with mechanosensors in their legs (McKelvey *et al.*, 2021), and rendering these sensors insensitive to vibrations via genetic manipulations reduces the mating success of a male. There currently does not exist physiological data on whether the leg mechanosensors do indeed respond to the very weak vibrations, so this remains to be tested, but we consider this sufficient evidence for calling vibrations a signal.

Given the behavioral evidence that both song and vibration are produced exclusively in a social context and that they affect female behaviors, we believe calling them "signals" is justified. In addition, "signal" is also a more specific term than "action," since it implies something being transmitted through a medium via the air (song) or the substrate (vibration).

We have expanded the relevant sentence in the introduction, clearly stating that more evidence is needed for vibrations (p2, l51):

Both song and vibration influence female mating behaviors and can therefore be considered signals (Stegmann 2013). In receptive females, song elicits acceptance behaviors such as slowing and vaginal plate opening (Deutsch 2019, Wang 2021). Conversely, unreceptive females display rejection behaviors in response to song, including acceleration and ovipositor extrusion (Coen 2014, Mezzera 2020). Although the evidence for vibrations is less conclusive and requires further investigation, several studies indicate that vibrations also elicit female acceptance behaviors, including slowing and copulation (Fabre 2012, McKelvey 2021), suggesting that vibration similarly functions as a signal.

Despite evidence that both signals affect female behavior, how the male brain coordinates air-borne song and substrate-borne vibration remains unknown.

2. In Fig 1B-E, the authors identify three behavioral states: sine, pulse, and vibrations. In F and G, however, the authors also plot the probability of songs (0.14) independently and the total probability adds up to 1.14. This is technically wrong and potentially confusing. Since sine + pulse probabilities also add up to be 0.14, I believe the authors simply combined the pulse and sine modes to define the song. The authors might want to clarify this issue and either plot sine vs. pulse vs. vibration or song vs. vibration.

Indeed, $p(\text{song})=p(\text{sine}) + p(\text{pulse})$. To avoid confusion, we removed $p(\text{song})$ and now only show $p(\text{sine})$, $p(\text{pulse})$ and $p(\text{vib})$ in panel 1F, as suggested.

Modified Fig. 1F:

3. In Fig. 2C, I am surprised and a bit worried that the true "no signal" category is incorrectly predicted to be a signal (song or vibration) majority of times (54%). This seems to be even higher for all three panels of Fig. 1E as well. Shouldn't "no signal" be the easiest state to classify? Does this not raise concerns about whether we can trust the GLM output for songs and vibrations?

We thank the reviewer for raising this important point. We were also initially concerned by this finding and had conducted additional analyses to address it. While the model's overall accuracy may appear modest, it correctly predicts when males sing or vibrate, 68% and 83% of the time. This demonstrates that the model identifies the behavioral contexts in which song and vibration are produced with high accuracy. Most errors made by the GLM are false positives: Song or vibration are often predicted even when the male produces no signal. We interpret this as the GLM successfully identifying contexts in which signals are produced, but that stochasticity or hidden internal states ultimately determine whether the male will actually produce signals in each instance. Although methods exist for identifying state-dependent behaviors (Calhoun 2019), these models are challenging to fit and interpret. We therefore opted to validate our modeling results through complementary behavioral analyses and causal manipulations. First, we confirmed that the contexts identified by our GLMs to predict vibration - such as far distance and slow female movement - are indeed significantly associated with vibrations (Fig. 2I, S3C-D). Second, we performed direct causal tests by experimentally manipulating female locomotion, demonstrating that female stationarity and movement do indeed control vibration production (Fig. 3). Note, even in these controlled experiments, stopping the female induced vibrations with only 40% probability, supporting our interpretation that additional stochastic or state-dependent factors influence the male's final decision to vibrate.

To address the false positive errors, we added the following sentence to Results (p4, I108, new text in bold italics):

*A model fitted using all 19 cues predicted the male's choice to vibrate with only few confusions (83% correct), demonstrating that vibrations, just like song, are produced not randomly but in a context-dependent manner (Fig. 2C). **Most errors were false positives (predicted song or vibration during "no signal"), implying that additional factors, such as stochasticity or internal states, further contribute to the male's signal choice (Calhoun 2019).***

4. In line 105, the authors write that "With only rare confusions between song and vibration we were able to determine that feedback cues determine the choice between song and vibration (Fig. 2C)." How is Fig. 2C showing that feedback cues are important?

The GLMs predicted vibrations with high accuracy (83%) from the tracking data, which we use as proxies for the male's locomotor state and the female behavior (e.g. her locomotion). This implies that vibrations are, just like song, not produced randomly, but "chosen" by the male in a context-dependent manner.

The Results section has been rephrased to provide greater precision on this matter.

(P4, I102):

*OLD: The choice between sine and pulse song is based on **female feedback** and our analyses of the transitions between song and vibration suggest that this might also be true for vibrations.*

*NEW: The choice between sine and pulse song is based on **male locomotor state and female behavior** and our analyses of the transitions between song and vibration suggest that this might also be true for vibrations.*

(P4, I108):

OLD: With only rare confusions between song and vibration we were able to determine that feedback cues determine the choice between song and vibration.

NEW: A model fitted using all 19 cues predicted the male's choice to vibrate with only few confusions (83% correct), demonstrating that vibrations, just like song, are produced not randomly but in a context-dependent manner (Fig. 2C).

5. Since Fig. 2E plots accuracy of predictions, calling it a confusion matrix in the legend is a bit confusing.

Fig. 2E does show confusion matrices, not accuracy values. The confusion matrices in Fig. 2E come from models that were fitted to predict vibration, song or no signal from *individual cues* (distance only, female velocity only, male lateral velocity only). The accuracy values (%correct) for these and other models are shown in Fig. 2C. The confusion matrix in Fig. 2B comes from a model that was fitted to predict signaling using all 19 cues at once.

We changed the wording in the results (see also our reply to point 4 above) and the caption of Figure 2 to clarify this (p6):

D [...] Predictive performance (% correct) of individual male (blue), female (pink), and relative (yellow) cues. **Models were fitted to predict male signal choice using individual cues only. [...]**

E Confusion matrices for predicting the male's signal choice (V - vibration, S - song, N - no signal) using **the most predictive individual male cue** (lateral velocity, bottom), female cue (female velocity, middle), and relative cue (distance, top).

6. In Line 111, the authors write "These findings indicate that male and female movement patterns are the strongest predictors of vibration production during courtship." What could have been a viable alternative hypothesis that is ruled out here?

The GLMs were fitted to predict male vibration based on male and female locomotion, their distance, and their relative orientation. Thus, plausible alternative results could have been that males vibrate at a fixed distance or at a defined relative angle to the female (e.g., in front of the female). While distance is indeed predictive of vibrations, it is much less predictive than locomotor cues.

These alternatives are now stated in Results, p4, l117, new text in bold):

*These findings indicate that male and female locomotion, **rather than their distance or angle**, are the strongest determinants of vibrations.*

7. In Line 133, the authors write "Having access to vibrations during courtship, we found that part of this 'passive' 133 state is not idle, but that the male actively signals to the female". Similar to point 1 above, how do we know whether it is an advertisement signal.

As pointed out in our reply to point 1, we believe there is sufficient evidence that vibrations are like song a signal that affects female behavior. Please refer to our answer to point 1 above.

8. Since GLMs can only identify correlations between behavioral variables, I applaud the authors for manipulating female locomotion state to infer its causal effect on male's behavior. In line 158, the authors write "It is therefore likely that the male's locomotor state controls the choice between song and vibration, and is not influenced by the female movement." I agree with the authors but isn't this contradictory to the main message of Fig 3, whose caption says "Female immobility is a necessary and sufficient trigger for male vibrations." The authors should clarify their conclusion for this section.

We thank the reviewer for pointing out this contradiction. We've edited the caption to Fig. 2 to clarify our conclusion:

(P6, Figure caption):

OLD: Female immobility is a necessary and sufficient trigger for male vibrations.

NEW: Immobility is a necessary and sufficient trigger for male vibrations.

9. In Figure 4A, it will be helpful to include the pMP2 neuron as well.

Thank you for pointing this omission out. The pMP2 neuron is now included in Fig. 4A

In line 199: the authors write "Interestingly, at the offset of strong activation, we observed vibrations lasting 5–10 s 200 (Fig. 4F). pC2l neurons are known to produce sine song at activation offset [7, 10, 30] but this sine song is much shorter." So, is it pulse -> sine -> vibration? What fraction of times does the pulse state transition to sine en route to vibrations?

This is an interesting idea! We do find sine song at the end of and immediately after optogenetic activation of pC2I. However looking at the single-trial raster plot (old panel Fig. S5C attached below), we do not see frequent pulse-sine-vibration sequences.

We confirmed this visual impression using a statistical analysis of signal sequences at the offset of pC2I activation (See new Fig. S5I and J attached below):

Whereas 93% of males produced pulse during pC2I activation, only 18% switched to sine during or after activation. Out of those 18%, most switched to pulse (57%) or ceased signaling (32%) whereas only a small fraction vibrated afterwards (12%, ~2% of the total). Only 11 % of the males that actually vibrated upon pC2I activation sang sine song before, however all of these sine trains were preceded by pulse song.

Taken together, the pulse-sine-vibration sequence is rare. This is consistent with pulse-sine transitions being controlled by pIP10 and rebounds in the TN1 neurons of the male VNC (Roemschied 2023). We have shown that vibrations are not driven by pIP10 activation (Fig. 4B) but via a different mechanism. See our reply to point 11 below.

We now show transitions between the first three signal types at the onset and the offset of P1a and pC2I activation in new Figs S5G-J reproduced below and we added the following to Results (p7, l207):

In addition, pulse-sine-vibration sequences were rare and most transitions into vibrations were preceded by pulse song (Fig. S5G-J).

New Fig. S5G-J

Caption:

G–J Signal sequences produced after the onset (G, I) and offset (H, J) of optogenetic activation of P1a (G, H) and pC2l (I, J). The Sankey diagrams show the transitions between the first three signal types produced (N - no signal, grey; P - pulse song, orange; S - sine song, blue; V - vibration, green). The width of connecting bands is proportional to the transition probability between pairs of signals.

The procedure for analysing the signal sequences is now added to Method (p16, l522): *Signal sequences were analyzed in optogenetically manipulated solitary males by pooling trials across all activation intensities. Starting from stimulus onset or stimulus offset, we determined which signal the males started with and which two signal types followed. Pauses between signal trains lasting longer than 0.5 s were considered "no signal".*

10. In line 220, the authors write that "60% of the stationary males vibrated independent of activation strength (Fig. 4J), suggesting that P1a neurons do not induce a drive to vibrate which in turn stops males." This is clever and this kind of probabilistic transition probabilities will be helpful for the sine, pulse, and vibrational modes as well. The authors should consider showing the transition probabilities between all of these states for the optogenetic manipulation experiments.

We have added new Figures showing the dependence of song and vibration on locomotion for activation of P1a (newFigs S6A, B) and pC2l (new Fig. S6C).

Figure S6: Effect of neuronal activation on locomotion and signaling.

A Left: Male velocity before (B), during (D), and after (A) optogenetic activation of P1a. Dots correspond to trials, lines connect the medians for each epoch. Middle/right: Velocity of males that vibrating (green, V), non-vibrating males (black, nV), singing (red, S), and non-singing (NS, black) males during (middle) and after (right) activation (27 mW/cm², N=13 flies, 7 trials/fly).

B Same as A but for stronger optogenetic activation of P1a (209 mW/cm², N=3 flies, 7 trials/fly).

C Same as A but for optogenetic activation of pC2l in solitary males (83 mW/cm², N=6 flies, 7 trials/fly).

11. Given that there is known mutual inhibition between the sine and the pulse modes within the "song" state, situating the circuit for mutual inhibition between songs and vibrations within that context will be very helpful.

We thank the reviewer for the opportunity to clarify this.

As mentioned by the reviewer, a previous study has shown that mutual inhibition between the pulse and sine song modes is implemented in the VNC, in the TN1 neurons (Roemschied 2023). Our results show that mutual inhibition between song and vibration is implemented downstream of P1a and pC2l, either in the brain, for instance at the level of descending neurons, or in the VNC. This mechanism is likely independent of the one for switching between pulse and sine: For instance, pulse and sine often follow each other without any pause in a song bout. By contrast, song and vibration are typically separated by 1-second

long pauses, suggesting that the pulse-sine and song-vibration mutual inhibition circuits act with very different timing and dynamics.

We have added the following statement to Results to relate our results to those of Roemschied et al. (p9, l253):

*These results show that mutual inhibition reduces the overlap between multimodal signals in *Drosophila*. **Mutual inhibition also coordinates the rapid switching between pulse and sine song (Roemschied 2023), suggesting that similar circuit principles coordinate signal production across timescales.***

We have also added the following statement to Discussion (p13, l398):

Notably, the same circuit motif coordinates song production and is likely implemented via pulse- and sine-specific TN1 neurons that inhibit each other in the VNC (Roemschied 2023).

12. In Fig. 4E, about half of the magenta (song) trials are followed by green (vibration). This is close to 50% transition probability. In natural behavior (Fig 1), this transition is very rare. The authors discuss the potential circuit state differences between natural behavior vs. artificial activation with optogenetics. One additional confound is the potential off-target effects of acute perturbations (Otchy et al, 2015). Driving neural activity beyond the natural regime in recurrent circuits might overestimate causal effects. What do the authors think of this possibility? What is known about the neural activity of P1a neurons during vibrations? Considering potential off-target effects of artificial stimulation is a very interesting point. Unfortunately, very little is known about the activity of P1 during natural courtship, but activation will likely be more intermittent and weaker than our optogenetic activation stimulus. This strong and consistent activation likely unmasks circuit dynamics that are typically overwritten by the dynamics of external stimuli coming from the female and we think this is why we do not see frequent song-vibration transitions during natural courtship.

We have edited the relevant part in Results (p9, l263):

*However, signal dynamics during natural courtship with a female are much more variable (Fig. 1). For instance, the pulse to vibration transitions produced by pC2l activation (Fig. 4E) are rarely seen during natural courtship (Fig. 1l). **While we cannot rule out that consistent optogenetic activation of P1a and pC2l drives the circuit into a non-typical dynamical regime (Otchy 2015, Jazayeri 2017), we believe the differences in signal dynamics between optogenetic activation and natural courtship arise because P1a and pC2l are activated by intermittent and dynamical social cues from the female.** Specifically, P1a responds to contact and volatile pheromones (Clowney 2015, Kallman 2015, Zhang 2018), while pC2l responds to acoustic and visual cues (Deutsch 2019, Kohatsu 2015, Roemschied 2023).*

13. In Fig 5A-B, the blue inhibitory connection is hypothesized. Please highlight this clearly in the figure panel to avoid confusion.

We've now indicated the hypothesized inhibitory connection with a question mark in the Figure.

14. In Fig 5K-N, consider staggering the "control" vs. "satiated" data points. Currently, it is difficult to discern.

We now make the “control” and “satiated” data points more distinct by labeling them with “C” and “S” in panels 5K-N.

15. In line 279, the authors claim "we now show that pC2l neurons were also subject to motivational control implying a global effect of motivation on the courtship circuit." I am not convinced of this claim. The difference between the Song probabilities in satiation vs. control (purple lines vs. purple shaded area) upon pC2l during the actual stimulation epoch is quite small. The major difference between the two contexts is observed during stimulation offset. But couldn't offset responses be indirectly due to the context-dependence of P1a neuron or the RNN or the walking drive circuit?

We agree that the main effect of satiation is observed after stimulation and that this could also be explained by mechanisms independent of pC2l. We clarified this in Results (p11, I292):

OLD: we now show that pC2l neurons were also subject to motivational control implying a global effect of motivation on the courtship circuit.

NEW: we now show that satiation globally reduces the persistence of signaling during courtship implying that both song and vibration are under common motivational control.

16. In line 301, authors write that "Activation of the RNN requires strong and direct activation of P1a neurons and mediates the long-term persistence of multimodal signals via connections to the descending neurons for song and vibration." Is this known from experiments or is this an assumption of the model? In general, this paragraph may benefit from clarifying which of the model characteristics are derived from previous experimental facts vs. hypothesized to be the case.

We thank the reviewer for pointing this out. The fact that the RNN requires strong activation and acts via connections to descending neurons are indeed assumptions of the model. We've edited the paragraph to more clearly tag what are model assumptions and what is known.

(P11, I304):

*Second, all indirect effects of optogenetic activation---the vibrations at the offset of pC2l neuron activation as well as the persistent song and vibration after P1a activation---were mediated by P1a neurons **in our model**.*

(P11, I312)

***In our model**, pC2l activation induces slowly decaying activity in P1a neurons, but is unable to engage the RNN downstream of P1a neurons. **We implemented this by requiring the RNN to receive strong P1a activation for persistent activity**. The RNN in turn mediates the persistent production of multimodal signals through connections to descending neurons that control song and vibration.*

Reviewer #3 (Remarks to the Author):

This is an exciting and important work which expands our understanding of Drosophila courtship behavior, and identifies neural circuit motifs for the integration of multimodal behaviors. The authors develop a novel behavioral assay to simultaneously quantify

vibrational and acoustic (song) communication occurring during courtship. Precise behavioral analysis reveals that the absence of locomotion, stationarity, is highly coordinated with vibration while movement is highly correlated with song. Behavioral overlap between vibrational and acoustic cues is rarely observed which suggested these behaviors could be coordinated by a single circuit.

Through a series of optogenetic experiments the authors identify separate premotor pathways responsible for both song (mediated through PC2I) and vibration (mediated through P1a) and evidence for mutual inhibition between the two pathways downstream of P1a and PC2I. In addition, they provide evidence descending pathways from P1a can also coordinate vibration and location. Based on these results the authors propose several models which they then elegantly test with additional behavioral and optogenetic manipulations. Ultimately, this work provides a compelling circuit dissection of multimodal elements of courtship behavior with elements downstream of P1a driving vibration and elements downstream of PC2I, driving song. The authors further demonstrate that coordination between the two behaviors can be explained by mutual inhibition. These results are supported by natural courtship behaviors, optogenetic manipulations and modeling.

The question is of broad interest, the methods are innovative and the conclusions are well supported by the data, making the work well suited to publication in Nature Neuroscience.

Major comments:

- Did the authors test if any features of the vibration behaviors, e.g. timing, duration, onset are influenced by the substrate? As this is an artificial setting, I would be curious to see how vibrations may be altered if the substrate was more similar to conditions of natural courtship (e.g. fermenting organic matter or fruit). While this is not necessary to reproduce here, it would be interesting to see how stereotyped the behavior is if the transmission of the vibration is changed by the substrate.*

How the substrate affects vibration behaviors is indeed an interesting question and has been tested before. McKelvey *et al.* (2021) have used laser vibrometry to record vibrations produced on banana, apple, prickly-pear cactus fruit, and reflective foil. They find only weak effects of substrate on vibration parameters, with vibration intervals ranging between 165 - 260 ms and vibration peak frequencies ranging between 300 - 500 Hz across substrates. Vibrations on apple and banana were produced at intervals of 200 ms, on prickly pear at intervals of 250 ms. Thus, it is possible that proprioceptive feedback modulates vibration production. The vibration recorded using paper by us falls into a similar range (IVI 150-200 ms and peak frequency ~300 Hz). McKelvey *et al.* also show that on different natural (apple, banana, cactus fruit) and artificial substrates (reflective foil membrane) males vibrate equally frequently. This suggests that the tendency to vibrate is independent of substrate consistent with our results that vibrations are driven by male and female stationarity.

Minor comments:

- *Methods: Could the authors please comment on how the LED intensities for optogenetic activation were chosen when only a single intensity is used.*

We have added that information to the Methods section (p16, l500):

For pC2l and P1a activation (Fig.4-5) we used LED intensities 14, 27, 83, and 209 (P1a only)mW/cm². In subsequent experiments for which only a single intensity was used, we selected the lowest intensity that evoked reliable vibrations (27 mW2/cm² for pC2l and P1a). Each experiment consisted of 7 trials of optogenetic stimulation and each trial started with 5 s of optogenetic stimulation followed by a pause of 120 s.

And (P16, l491):

This intensity was chosen in pilot experiments to induce reliable female stationarity while minimizing off-target effects from the light stimulus in male courters.

- *Figure 5C,D: The schematic of the courting flies seems a bit misaligned and distracts from the figure. Please reposition the schematic or consider removing it.*

We removed the schematic of the courting flies.

- *Figure 6A: PC2l is missing the l.*

Fixed.

- *Figure 6 legend: In the first sentence a space is missing between "excitatory and inhibitory"*

Fixed.

- *Figure S9 is embedded in the middle of the references.*

Fixed.

REVIEWERS' COMMENTS

Reviewer #1 (Remarks to the Author):

The authors have done an excellent job of rewriting the abstract and addressing all of my minor concerns. I do not have any further comments and strongly support the acceptance of this manuscript for publication in Nature Communications. This is an excellent piece of work.

Reviewer #2 (Remarks to the Author):

The authors have done a wonderful job in addressing my concerns. I support the publication of the manuscript in Nature Communication.

Reviewer #3 (Remarks to the Author):

My comments have been addressed and I thank the authors for the detailed responses. I find the manuscript much improved by the author's response to all Reviewers and recommend the manuscript for publication.

We thank the reviewers for their positive comments.